# Make Memory Buffer Stronger in Continual Learning: A Continuous Neural Transformation Approach

## Abstract

Continual learning (CL) focuses on learning non-stationary data distribution without forgetting previous knowledge. However, the most widely used memory-replay approach often suffers from memory overfitting. To mitigate the memory overfitting, we propose a continuous and reversible memory transformation method so that the memory data is hard to overfit, thus improving generalization. The transformation is achieved by optimizing a bi-level optimization objective that jointly learns the CL model and memory transformer. Specifically, we propose a deterministic continuous memory transformer (DCMT) modeled by an ordinary differential equation, allowing for infinite memory transformation and generating diverse and hard memory data. Furthermore, we inject uncertainty into the transformation function and propose a stochastic continuous memory transformer (SCMT) modeled by a stochastic differential equation, which substantially enhances the diversity of the transformed memory buffer. The proposed neural transformation approaches have significant advantages over existing ones: (1) we can obtain infinite many transformed data, thus significantly increasing the memory buffer diversity; (2) the proposed continuous transformations are reversible, i.e., the original raw memory data could be restored from the transformed memory data without the need to make a replica of the memory data. Extensive experiments on both task-aware and task-free CL show significant improvement with our approach compared to strong baselines.

## 1 Introduction

Continual learning (CL) aims to learn non-stationary data distribution without forgetting previous knowledge. Depending on whether there are explicit task definitions (partitions) during training, CL can be categorized into task-aware and task-free CL. For task-aware CL, there are explicit tasks and class splits during training; according to whether the task identities are known or not during testing, it can be further categorized into task/domain/class-incremental CL (van de Ven & Tolias, 2019). For task-free CL (Aljundi et al., 2019b), there is no explicit task definition, and data distribution shift could happen at any time.

Memory replay is an effective way to mitigate forgetting and has been widely used in CL. One major problem of the memory-based methods is that the effectiveness of memory buffer data could gradually decay during training (Delange et al., 2021; Jin et al., 2021), i.e., the CL model may overfit the limited memory data and could not generalize well to the previous tasks. Recently, gradient-based memory editing (GMED) (Jin et al., 2021) has been proposed to mitigate memory overfitting by editing memory data with *hard* examples in a way similar to adversarial data augmentation (ADA) (Madry et al., 2018). Specifically, it creates hard examples that increase model losses at each gradient step but restricts to a few (less than three) *discrete* gradient-based editing steps. With more editing steps, similar to ADA, GMED would make the memory even harder but cause less data diversity since the adversarial force will drive the feature space of different classes overlap and cluster together (Madry et al., 2018; Wang et al., 2021). However, as studied in previous work (Gontijo-Lopes et al., 2021), improving the diversity of training data is crucial to improving model generalization. An illustration of this phenomenon is shown in Figure 1 (b) and (c). This naturally leads to a new

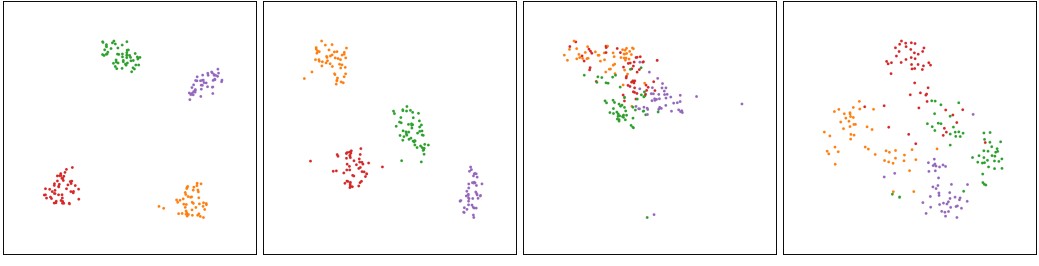

(a) experience replay (ER)   (b) GMED (3 edit steps)   (c) GMED (10 edit steps)   (d) DCMT(Ours)

Figure 1: T-SNE visualization of existing memory-replay and proposed methods on CIFAR10. We use features extracted from the last layer output of ResNet18 as the input to T-SNE. We use four classes of memory data to illustrate the difference. T-SNE embeds each data point, and each color denotes one class of memory buffer data. (a): ER is very easy to overfit and easy to classify; (b): GMED with smaller editing steps has limited effectiveness due to the limited hardness; (c): GMED with larger editing steps creates memory examples harder to classify but lack of diversity; (d): DCMT (our method) with better diversity and transformed memory data is hard to classify and overfit.

problem: *how can we increase the diversity of the edited memory data and maintain its hardness at the same time?*

To address this problem, we present a continuous, expressive, and flexible memory transformation method to obtain a diverse set of memory data and make the memory buffer harder to memorize at the same time. We first model the gradual and continuous memory transformation as a deterministic neural ordinary differential equation in the time interval $[0, T]$, named Deterministic Continuous Memory Transformer (DCMT). There are several advantages compared to existing methods. First, we can obtain infinite time steps of transformed memory data for any $t \in [0, T]$ and thus significantly improve the diversity in the transformed memory data. Second, we do not need to make a replica of the raw memory data since the transformation process is reversible. We can restore original raw memory data from the transformed memory data. As shown in Figure 1(d), DCMT diversifies memory data while maintaining hardness.

The proposed DCMT considers a *single* transformation function. However, there are infinite possible transformation functions for transforming the memory data, and it is beneficial to model the uncertainty in the transformation function to further avoid overfitting (Lu et al.; Liu et al., 2019). To model the underlying various transformation functions and further improve the data diversity, we thus generalize the methods in a probabilistic manner to model the memory transformation as a stochastic process with neural stochastic differential equations, named Stochastic Continuous Memory Transformer (SCMT). This enables us to model infinite transformation functions and significantly improves the diversity of the transformed data with some increased computation cost compared to DCMT. The overview of the proposed methods is presented in Figure 2.

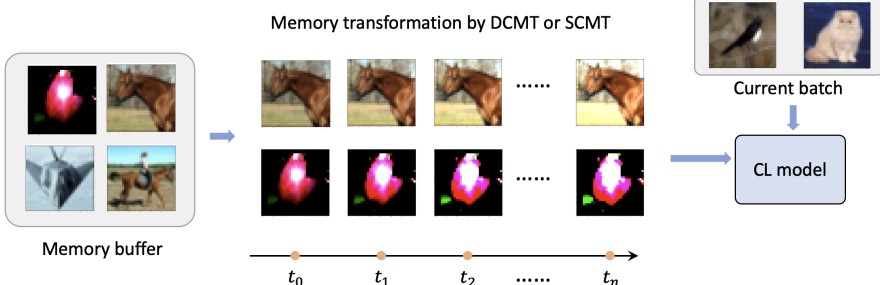

Figure 2: Overview of the proposed approach for memory transformation. DCMT and SCMT continuously and gradually transform the memory data to be diverse and hard to memorize. Note that the transformed data could be obtained at any *continuous time step*, thus providing significantly larger diversity.

We propose a bi-level optimization to jointly learn the memory transformer and CL model. The memory transformer can generate diversified memory data that is hard to memorize. Concretely, after continuous interval $[0, T]$ transformation, we optimize the loss increase before ($t = 0$) and after

$(t = T)$ the transformation to ensure the hardness of the transformed data. To ensure that the network embeds similarly for both the original raw memory data and the transformed memory data, we adopt a Jensen-Shannon divergence consistency loss to regularize the network output to ensure smoother neural network responses for both data. Furthermore, our proposed method is general, versatile, and can be seamlessly applied to both task-aware and task-free CL. Extensive experiments on both task-aware and task-free CL demonstrate the effectiveness of the proposed methods. We summarize our contributions as three-fold:

- We propose a bi-level optimization framework with a continuous memory transformer to address memory overfitting issues. The continuous memory transformer can make the transformed memory data substantially more diverse and harder to memorize.
- We instantiate the continuous memory transformation with deterministic and stochastic memory transformation, which can be seamlessly applied to both task-aware and task-free CL.
- We perform extensive experiments on both task-aware and task-free CL, showing significant improvements compared to strong baselines. Furthermore, we provide detailed ablation studies.

## 2 PRELIMINARY AND RELATED WORK

### 2.1 TASK-AWARE CL

**Problem setup.** Task-aware CL focuses on the case where there are explicit task definitions during CL. Task/domain/class-incremental learning (van de Ven & Tolias, 2019) are the three most representative CL scenarios. We consider the problem of learning a sequence of tasks denoted as $\mathcal{D}^{tr} = \{\mathcal{D}_1^{tr}, \mathcal{D}_2^{tr}, \cdots, \mathcal{D}_N^{tr}\}$, where $N$ is the number of training tasks. The $k$-th task training data $\mathcal{D}_k^{tr}$ consists of a set of triplets $\{(\boldsymbol{x}_i^k, y_i^k, \mathcal{T}_k)_{i=1}^{n_k}\}$, where $\boldsymbol{x}_i^k$ is the $i$-th data example in the task, $y_i^k$ is the corresponding data label, and $\mathcal{T}_k$ is the task identifier. The goal is to learn a model $f_{\boldsymbol{\theta}}$ on the training task sequence $\mathcal{D}^{tr}$ so that it performs well on the test set of all the learned tasks $\mathcal{D}^{te} = \{\mathcal{D}_1^{te}, \mathcal{D}_2^{te}, \cdots, \mathcal{D}_N^{te}\}$ without forgetting previously learned knowledge.

**Existing work.** The proposed approaches for task-aware CL can be categorized into: 1) maintaining a memory buffer that stores previous examples for future replay (Lopez-Paz & Ranzato, 2017; Shin et al., 2017; Chaudhry et al., 2019a; Riemer et al., 2019; Chaudhry et al., 2019b; Aljundi et al., 2019a; PourKeshavarzi et al., 2022; Arani et al., 2022); 2) using dynamic network architectures (Rusu et al., 2016; Fernando et al., 2017; Yoon et al., 2018; Qin et al., 2021; Miao et al., 2022) and remembering past knowledge by dynamically updated architectures; 3) enforcing regularization to slow down forgetting (Kirkpatrick et al., 2017; Zenke et al., 2017b; von Oswald et al., 2020; Liu & Liu, 2022; Raghavan & Balaprakash, 2021); and 4) modeling the parameter update uncertainty with Bayesian methods (Nguyen et al., 2018; Ebrahimi et al., 2020; Henning et al., 2021). In this paper, we focus on memory-replay-based methods since they often achieve SOTA performance.

### 2.2 TASK-FREE CL

**Problem setup.** Task-free CL (He et al., 2019; Zeno et al., 2019; Aljundi et al., 2019b; Chrysakis & Moens, 2020; Lee et al., 2020) is a recent generalization of CL to the more complex cases, where data distribution shift could happen at any time during CL without explicit definition of tasks.

**Existing work.** Most existing works in task-free CL are memory-replay-based methods (Chaudhry et al., 2019b;a). Our works share a similar goal with GMED (Jin et al., 2021) which edits the memory buffer based on ADA, making the memory data harder but lacks diversity (Madry et al., 2018; Wang et al., 2021). There are several significant differences. First, GMED is a *gradient-based discrete step* memory editing method. Our memory transformer can obtain *infinite continuous time steps* transformation of memory data, which improves the memory diversity significantly. Second, GMED overwrites the memory data with the edited ones making the memory buffer data distribution significantly deviate from the original raw memory data distribution after many epochs of editing, especially in task-aware CL, which would decrease the performance. In contrast, our transformation process is reversible, i.e., the original raw data can be recovered from the transformed data. Thus, we do not need to overwrite the memory buffer data or keep an additional mini-batch transformed data.

We provide more detailed discussions of related work in Appendix C due to space limitations.

## 3 METHODOLOGY

In this section, we first present standard memory replay of CL in Section 3.1, our proposed deterministic continuous memory transformer (DCMT) in Section 3.3 and stochastic continuous memory transformer (SCMT) in Section 3.4. Then, we present the training objectives in Section 3.5. The overall description of the proposed method is shown in Figure 2.

### 3.1 CONVENTIONAL MEMORY REPLAY

Standard memory replay for CL (Chaudhry et al., 2019b) is to optimize a risk of data from both memory buffer $\mathcal{M}$ and current mini-batch. Formally speaking, the optimization can be formulated as:

$$\min_{\forall \boldsymbol{\theta} \in \boldsymbol{\Theta}} \left[ \mathcal{L}(\boldsymbol{\theta}, \boldsymbol{x}_k, y_k) + \mathbb{E}_{(\boldsymbol{x}, y) \sim \mathcal{M}} \mathcal{L}(\boldsymbol{\theta}, \boldsymbol{x}, y) \right], \tag{1}$$

where $k$ is the CL timestamp ($k^{th}$ CL step), $\boldsymbol{\theta}$ are model parameters, and $\mathcal{L}(\boldsymbol{\theta}, \boldsymbol{x}, y)$ is the loss function associated with the data $(\boldsymbol{x}, y)$. Conventional memory replay would make the memory buffer data gradually become less effective for mitigating forgetting when training for a long time, as it is easy to overfit the limited memory buffer data (Delange et al., 2021; Jin et al., 2021). Thus, the previously learned knowledge would get lost, and the CL model may not generalize well to the previous tasks. We thus propose a continuous memory transformation method to generate diversified memory buffer data that is hard to memorize in the following sections.

### 3.2 A PRELIMINARY APPROACH TO INCREASE MEMORY DIVERSITY

A preliminary way to increase the memory diversity is to transform the memory data with a neural network function $g$ parameterized by $\boldsymbol{\phi}$. A sequence of transformations can be applied on the original raw memory data $\boldsymbol{x}^m(t)$ by :

$$\boldsymbol{x}^m(t + (i+1)\Delta_t) = \boldsymbol{x}^m(t + i\Delta_t) + g(\boldsymbol{x}^m(t + i\Delta_t), t, \boldsymbol{\phi})\Delta_t, \quad i = 0, 1, \cdots, n \tag{2}$$

where $\Delta_t$ is the step size. By repeating this discrete transformation process, we can obtain a diverse collection of transformed memory data. However, when we updating the function $g$ by backpropagation, we need to store all the intermediate transformations, i.e., $\{\boldsymbol{x}^m(t + i\Delta_t), i = 0, \cdots, n-1\}$. Thus, the memory cost scales linearly with the number of memory transformation steps, i.e., $\mathcal{O}(n)$. This would bring a lot of memory cost especially if we transform the memory data by a large number of transformation steps. Furthermore, the neural network function $g$ is generally not invertible. We thus also need to store both the raw memory data and the transformed data. In the following, we simultaneously addressed the above issues by viewing the memory transformation process as a continuous dynamic system. This brings several benefits: (1) we do not need to store any intermediate transformation results, i.e., the memory cost is constant, i.e., $\mathcal{O}(1)$; (2) we do not need to store both the raw memory data and transformed data since the entire transformation is invertible even if the function $g$ is not invertible (More elaboration on this is provided in Appendix B.13). But the above discrete transformation needs to do so since $g$ is generaly not invertible; (3) our method brings infinite amount of transformed memory data vs. the discrete steps of transformations. We name this our proposed preliminary method as **Discrete Transformation (DT)**.

### 3.3 MEMORY TRANSFORMER AS A DETERMINISTIC CONTINUOUS DYNAMIC SYSTEM

In this section, we first view the memory transformation as a deterministic dynamic system. We transform the raw memory data into a continuous system in the time interval $[0, T]$. Suppose at each CL timestamp $k$, and we sample a mini-batch data $(\boldsymbol{x}^m, \boldsymbol{y}^m)$ from the memory buffer $\mathcal{M}$. Since we perform similar continuous mini-batch memory transformation operations at each CL timestamp $k$, we thus omit $k$ for presentation clarity. We model the gradual and continuous memory data transformation by the following differential equation:

$$\frac{d\boldsymbol{x}^m(t)}{dt} = g(\boldsymbol{x}^m(t), t, \boldsymbol{\phi}), \quad \boldsymbol{x}^m(0) = \boldsymbol{x}^m \tag{3}$$

where the memory transformer is parametrized by function $g$ with parameters $\boldsymbol{\phi}$ and $g$ represents the instant time transformation rate of memory data. By integrating both parts of the Eq. (3) over the

time interval $[0, T]$, we can obtain the solution to Eq. (3) for the transformed memory data at time $T$:

$$\boldsymbol{x}^m(T) = \boldsymbol{x}^m(0) + \int_0^T g(\boldsymbol{x}^m(t), t, \boldsymbol{\phi})dt \tag{4}$$

where the transformed memory data at any time $T$, i.e., $\boldsymbol{x}^m(T)$ is a *continuous function* of $T$. For any $t \in [0, T]$, $\boldsymbol{x}^m(t)$ is a transformation of original memory data, thus we can obtain a set of infinite transformed memory data, i.e., $\{\boldsymbol{x}^m(t) : t \in [0, T]\}$. This is in contrast to GMED (Jin et al., 2021), which works with a small number (less than three) of discrete-time steps of memory editing. With longer editing steps, similar to ADA, the edited data examples by GMED become harder but decrease the data diversity; thus, performance drops (Wang et al., 2021). Therefore, the design principle restricts the expressiveness and effectiveness of GMED. However, our memory transformation is significantly more expressive and provides substantially more diverse transformed memory data than GMED. We name our method as **Deterministic Continuous Memory Transformer (DCMT)**. In practice, we can use a numerical integration scheme, such as the Runge-Kutta method (Schober et al., 2014), similar to the implementation in (Chen et al., 2018) to solve the Eq. (4). We provide the algorithm details in Algorithm 3 in Appendix B.12. The above transformation process is reversible because we can obtain the raw memory data by the following reverse integration:

$$\boldsymbol{x}^m(0) = \boldsymbol{x}^m(T) + \int_T^0 g(\boldsymbol{x}^m(t), t, \boldsymbol{\phi})dt. \tag{5}$$

Eq .5 transforms from $\boldsymbol{x}^m(T)$ into raw memory data $\boldsymbol{x}^m(0)$ by integrating over the *reverse time interval* $[T, 0]$. We can thus discard the raw memory data and only keep the transformed memory data. After the replay, we can invert the transformed memory data into original data. The transformation function $g$ does not need to be reversible and only needs to be uniformly Lipschitz continuous in $\boldsymbol{x}^m(t)$ and continuous in $t$ (This condition is to make sure the solution to Eq. (3) exits and is unique), thus providing great flexibility for the transformation function design (More elaboration on this is provided in Appendix B.13). Thus, our method has no extra memory cost to store additional raw memory data.

### 3.4 Memory transformer as a stochastic continuous dynamic system

The DCMT method in Section 3.3, i.e., Eq. (3), only considers a single deterministic transformation function $g$, but there are infinite possibilities of available memory transformation functions. Thus, a single deterministic transformation is insufficient for modeling the underlying high diversity in the memory transformation functions. Furthermore, (Lu et al.; Liu et al., 2019) show that adding uncertainty modeling for the network is beneficial to avoid overfitting.

We thus model the memory transformation process as a stochastic dynamic system with a path-valued random variable $X : [0, T] \to \mathbb{R}^d$, where each random variable at time $t$, i.e., $X_t$, is to model the distribution of the transformed memory data $\boldsymbol{x}^m(t)$ at time $t$. We use $d$ to denote the memory data dimension. Let $W : [0, T] \to \mathcal{W}^w$ be a $w$-dimensional Brownian motion (Øksendal, 2014) which is a continuous time stochastic process such that $W_{t+s} - W_s$ follows a Gaussian distribution with mean 0 and variance $t$. Let $\mu_{\boldsymbol{\phi}} : [0, T] \times \mathbb{R}^d \to \mathbb{R}^d$ be the network for modelling the drift term, and $\sigma_{\boldsymbol{\phi}} : [0, T] \times \mathbb{R}^d \to \mathbb{R}^{d \times w}$ be the network for modeling the diffusion term. They are parameterised together by $\boldsymbol{\phi}$. For notation and presentation clarity, we still use the same notations, $\boldsymbol{\phi}$, as DCMT to denote the parameters of memory transformer. The memory transformation stochastic process can be modeled as the following stochastic differential equations (SDE):

$$dX_t = \mu_{\boldsymbol{\phi}}(t, X_t)dt + \sigma_{\boldsymbol{\phi}}(t, X_t) \circ dW_t, \quad \boldsymbol{x}^m \sim X_0, \tag{6}$$

where the initial values of the SDE are the raw memory data that can be viewed as samples from the initial random variable $X_0$. For all $t \in [0, T]$, let $X : [0, T] \to \mathbb{R}^d$ denote the solution to Eq. (6) and $\circ dW_t$ denotes Stratonovich integration (defined in Appendix B.11). The stochastic process $\{X_t\}_{t \in [0, T]}$ determined by Eq. (6) can be equivalently expressed as following:

$$X_T = X_0 + \int_0^T \mu_{\boldsymbol{\phi}}(t, X_t)dt + \int_0^T \sigma_{\boldsymbol{\phi}}(t, X_t) \circ dW_t. \tag{7}$$

Each random variable $X_t$ of $\{X_t\}_{t \in [0, T]}$ models a transformed memory data distribution. Thus, we obtain infinite transformed memory data distributions. This is in contrast to DCMT, where each

$\boldsymbol{x}^m(t)$ is deterministic. To solve this SDE and achieve cheap backpropagation, we use the adjoint method, i.e., the reversible Heun method proposed in (Kidger et al., 2021), (algorithm details shown in Algorithm 5 in Appendix B.12.1). We name this method as **Stochastic Continuous Memory Transformer (SCMT)**. Therefore, similar to DCMT, SCMT does not need to make a replica of the raw memory data.

### 3.5 TRAINING OBJECTIVES FOR CONTINUOUS NEURAL MEMORY TRANSFORMER

The goal of the memory transformer is to make the memory data hard to be memorized for the CL model. Our overall learning objective is the following bi-level optimization:

$$\min_{\boldsymbol{\theta}} \left[ \mathcal{L}(\boldsymbol{x}_k, y_k, \boldsymbol{\theta}) + \mathcal{L}(\widehat{\boldsymbol{x}^m}(T), \boldsymbol{y}^m, \boldsymbol{\theta}, \boldsymbol{\phi}_*) \right] \tag{8}$$

$$s.t. \quad \boldsymbol{\phi}_* = \arg \max_{\boldsymbol{\phi}} \left[ \mathcal{L}(\boldsymbol{x}^m(T), \boldsymbol{y}^m, \boldsymbol{\theta}, \boldsymbol{\phi}) - \mathcal{L}(\boldsymbol{x}^m, \boldsymbol{y}^m, \boldsymbol{\theta}, \boldsymbol{\phi}) - \lambda \mathbb{JS}(\boldsymbol{x}^m, \boldsymbol{x}^m(T)) \right] \tag{9}$$

$$\text{where} \quad \boldsymbol{x}^m(T) = \boldsymbol{x}^m(0) + \int_0^T g(\boldsymbol{x}^m(t), t, \boldsymbol{\phi})dt, \ \widehat{\boldsymbol{x}^m}(T) = \boldsymbol{x}^m(0) + \int_0^T g(\boldsymbol{x}^m(t), t, \boldsymbol{\phi}_*)dt \tag{10}$$

where $\boldsymbol{x}^m(T)$ could be either from DCMT or samples from the terminal state of SCMT, $\boldsymbol{\phi}$ denotes the parameters of either DCMT or SCMT. The lower-level optimization, i.e., Eq. (9), is to make memory buffer data hard to be memorized, and $\boldsymbol{\phi}_*$ is the obtained optimal solution. The last term $\mathbb{JS}(\boldsymbol{x}^m, \boldsymbol{x}^m(T))$ is to ensure smoother model responses on the original raw and transformed data; where $\lambda$ is the regularization strength. We will discuss this in detail in the following. The upper-level optimization, i.e., Eq. (8), is to replay the transformed memory data. Note that $\widehat{\boldsymbol{x}^m}(T)$ is the transformed memory data by the optimal memory transformer with parameter $\boldsymbol{\phi}_*$, defined in Eq. 10 (right). The above bi-level optimization is for DCMT, but the method can also be directly applied to SCMT. In practice, besides the obtained $\widehat{\boldsymbol{x}^m}(T)$, we can obtain infinite time steps transformed data at any time in the interval $[0, T]$. To make the computation tractable, we can randomly sample time steps, i.e., $0 = t_0 < t_1 < t_2 \cdots t_n = T$ and obtain the transformed memory data $\widehat{\boldsymbol{x}^m}(t_0), \widehat{\boldsymbol{x}^m}(t_1), \widehat{\boldsymbol{x}^m}(t_2), \cdots, \widehat{\boldsymbol{x}^m}(T)$ without additional cost since they are already in the integration interval $[0, T]$, which can be used for memory replay. It is worth noting that the transformed data at different time stamps are not combined and they will not be added to the memory buffer. First, previous transformed data has already been learned by the CL learner. For new tasks, we need to transform the memory data adaptively. Second, storing those data would increase a lot of memory storage cost. To maintain the diversity of the transformed memory buffer and reduce computation cost, we randomly sample $n$ from [1, 5] at each CL step. Note that the number of parameters in DCMT or SCMT, i.e. $\boldsymbol{\phi}$, is much smaller than that of CL model, $\boldsymbol{\theta}$, thus negligible compared to CL model backbone.

**Consistency loss.** The memory transformer could generate diverse memory data. To ensure that the network embeds similarly for both the original raw memory data and the transformed memory data, we use a similar consistency loss (Hendrycks et al., 2020) to regularize the network output to ensure smoother CL model responses. The goal is to make the CL model respond similarly to $\boldsymbol{x}^m(T)$ and $\boldsymbol{x}^m$, thus minimize the Jensen-Shannon divergence among the posterior distributions of the original sample $\boldsymbol{x}^m$ and its transformed variants $\boldsymbol{x}^m(T)$. The consistency loss is defined as below:

$$\mathbb{JS}(\boldsymbol{x}^m, \boldsymbol{x}^m(T)) = (\mathbb{KL}(p_{\boldsymbol{x}^m}|p_{mean}) + \mathbb{KL}(p_{\boldsymbol{x}^m(T)}|p_{mean})))/2, \text{where} \ \ p_{mean} = (p_{\boldsymbol{x}^m} + p_{\boldsymbol{x}^m(T)})/2$$

where $\mathbb{KL}$ denotes the KL divergence between two distributions, $p_{\boldsymbol{x}^m} = f_{\boldsymbol{\theta}}(\boldsymbol{x}^m)$ is the network output probabilities of each class for original raw data $\boldsymbol{x}^m$. Similarly, we can define $p_{\boldsymbol{x}^m(T)} = f_{\boldsymbol{\theta}}(\boldsymbol{x}^m(T))$.

The CL model parameters are then updated using the transformed memory data and the mini-batch data received at timestamp $k$ as follows:

$$\boldsymbol{\theta}_{k+1} = \boldsymbol{\theta}_k - \eta \nabla_{\boldsymbol{\theta}}[\mathcal{L}(\boldsymbol{\theta}_k, \widehat{\boldsymbol{x}^m}(T), \boldsymbol{y}^m) + \mathcal{L}(\boldsymbol{\theta}_k, \boldsymbol{x}_k, y_k)]$$

where $\eta$ is the learning rate. The learning process alternates between updating the memory transformer parameters $\boldsymbol{\phi}$ (using adjoint method (Pontryagin et al., 1962) to update the parameters in DCMT or using the reversible Heun method (Kidger et al., 2021) to update the parameters in SCMT) and the CL model parameters $\boldsymbol{\theta}$. The complete memory transformation algorithm is shown in Algorithm 1.

---
**Algorithm 1** Continuous Memory Transformation.

---
1: **REQUIRE:** model parameters $\boldsymbol{\theta}$, memory transformer parameters $\boldsymbol{\phi}$, CL model learning rate $\eta$, memory transformer learning rate $\beta$; transformation time $T$ at each iteration, memory buffer $\mathcal{M}$; $K$ is the number of iterations during the training process for both task-aware and task-free CL.
2: **for** $k = 1$ to $K$ **do**
3:     a mini-batch data $(\boldsymbol{x}_k, y_k)$ arrives.
4:     sample mini-batch data from memory buffer, i.e., $(\boldsymbol{x}^m, y^m) \sim \mathcal{M}$
5:     $\boldsymbol{x}^m(T) = \text{ODEsolver}(\boldsymbol{x}^m, 0, T)$ (Eq. (4)) by Algorithm 3 in Appendix B.12
       or $\boldsymbol{x}^m(T) \sim X_T = \text{SDEsolver}(\boldsymbol{x}^m, 0, T)$ (Eq. (7)) by Algorithm 4 in Appendix B.12
6:     calculate loss function $\mathcal{L}(\boldsymbol{\theta}, \boldsymbol{\phi}) = [\mathcal{L}(\boldsymbol{x}^m(T), \boldsymbol{y}^m, \boldsymbol{\theta}, \boldsymbol{\phi}) - \mathcal{L}(\boldsymbol{x}^m, \boldsymbol{y}^m, \boldsymbol{\theta}, \boldsymbol{\phi}) - \lambda \mathbb{JS}(\boldsymbol{x}^m, \boldsymbol{x}^m(T))]$
7:     we calculate the gradient $\frac{\partial \mathcal{L}(\boldsymbol{\theta}, \boldsymbol{\phi})}{\partial \boldsymbol{\phi}}$ with details provided in Appendix B.3 and Algorithm 2.
8:     update memory transformer parameters $\boldsymbol{\phi}$ by gradient ascent to maximize $\max_{\boldsymbol{\phi}} \mathcal{L}(\boldsymbol{\theta}, \boldsymbol{\phi})$
9:     transform memory data at randomly sampled time steps $0 = t_0 < t_1 < t_2 \cdots t_n = T$, i.e. $\widehat{\boldsymbol{x}^m}(t_i)$ by Eq. 10, for memory replay.
10:     update CL model parameters $\boldsymbol{\theta}$: $\boldsymbol{\theta}_{k+1} = \boldsymbol{\theta}_k - \eta \nabla_{\boldsymbol{\theta}} [\mathcal{L}(\boldsymbol{\theta}_k, \widehat{\boldsymbol{x}^m}(t_i), y) + \mathcal{L}(\boldsymbol{\theta}_k, \boldsymbol{x}_k, y_k)]$
11:     **if** restore **then**
12:         restore $\boldsymbol{x}_0^m$ with Eq. (5)
13:     **end if**
14:     update memory buffer by reservoir sampling (RS) (Vitter, 1985; Riemer et al., 2019), $\mathcal{M} = \text{RS}(\mathcal{M}, (\boldsymbol{x}_k, y_k))$
15: **end for**

---

# 4 EXPERIMENTS

We evaluate our methods for task-aware CL in Section 4.1 and task-free CL in Section 4.2.

**Datasets**. We compare different methods on **CIFAR10** (Krizhevsky, 2009) with 10 image classes, **CIFAR100** (Krizhevsky, 2009) with 100 image classes, **MinImageNet** (Vinyals et al., 2016) with 100 image classes and **Tiny-ImageNet** (Stanford, 2015) with 200 classes.

## 4.1 TASK-AWARE CL

We perform experiments on both task-incremental (Task-IL) and class-incremental (Class-IL) CL (van de Ven & Tolias, 2019). Task-IL provides task identities to the CL learner and is the easiest scenario. Class-IL does not provide task identities and is the hardest scenario in task-aware CL.

**Baseline**. We compare to various SOTA CL methods, including: 1) regularization-based methods, Classifier-Projection Regularization (CPR) (Cha et al., 2021), PASS (Zhu et al., 2021), Gradient Projection Memory (GPM) (Saha et al., 2021), oEWC (Schwarz et al., 2018), synaptic intelligence (SI) (Zenke et al., 2017a) and Learning without Forgetting (LwF) (Li & Hoiem, 2018); 2) Bayesian-based methods, UCB (Ebrahimi et al., 2020); 3) architecture-based methods, HAT (Serrá et al., 2018); 4) memory-based CL methods, including ER (Chaudhry et al., 2019b), A-GEM (Chaudhry et al., 2019a), GSS (Aljundi et al., 2019c), HAL(Chaudhry et al., 2021), DER++ (Buzzega et al., 2020) and GMED (Jin et al., 2021). The implementation for those baselines (Buzzega et al., 2020) already applies data augmentation, such as random crops and horizontal flips, etc. We apply the proposed method on top of those implementations. We adapt PASS(Zhu et al., 2021) to standard CL by adding noise to memory data. We provide detailed baseline descriptions in Appendix B.2.

**Evaluation metrics.** We evaluate the performance of the proposed methods and the compared methods with average accuracy (ACC) and backward transfer (BWT) at the end of CL training to measure the final performance and the degree of forgetting for different methods. We denote $a_{N,k}$ as the testing accuracy on task $k$ after learning on task $N$. The overall accuracy for all the tasks is $ACC = \frac{1}{N} \sum_{k=1}^{k=N} a_{N,k}$. To measure catastrophic forgetting, we also evaluate BWT, which measures the extent of forgetting on previous tasks after learning new ones. Formally, BWT is defined as: $BWT = \frac{1}{N-1} \sum_{k=1}^{k=N-1} (a_{N,k} - a_{k,k})$. $BWT < 0$ indicates forgetting of previous tasks, and $BWT > 0$ indicates that learning new tasks is helpful on previous tasks.

**Implementation details.** We follow (Buzzega et al., 2020) to use ResNet18 (He et al., 2016) as the classifier for all datasets. Following (Buzzega et al., 2020), we split the CIFAR-10 dataset into 5 disjoint tasks, where each task consists of 2 classes. We split MiniImagenet (Vinyals et al., 2016)

Table 1: **Task-IL and class-IL** results on CIFAR10, CIFAR-100 and MiniImagenet, respectively with memory size 500. '—' indicates not applicable.

| Algorithm | CIFAR-10 | | CIFAR-100 | | MiniImagenet | |
| Method | Class-IL | Task-IL | Class-IL | Task-IL | Class-IL | Task-IL |
|---|---|---|---|---|---|---|
| fine-tuning | $19.62 \pm 0.05$ | $61.02 \pm 3.33$ | $9.29 \pm 0.33$ | $33.78 \pm 0.42$ | $8.59 \pm 0.29$ | $27.48 \pm 0.38$ |
| Joint train | $92.20 \pm 0.15$ | $98.31 \pm 0.12$ | $71.32 \pm 0.21$ | $91.31 \pm 0.17$ | $65.56 \pm 0.18$ | $87.74 \pm 0.15$ |
| oEWC | $19.49 \pm 0.12$ | $68.29 \pm 3.92$ | $8.24 \pm 0.21$ | $21.2 \pm 2.08$ | $7.32 \pm 0.16$ | $22.54 \pm 1.78$ |
| SI | $19.48 \pm 0.17$ | $68.05 \pm 5.91$ | $9.41 \pm 0.24$ | $31.08 \pm 1.65$ | $8.07 \pm 0.26$ | $30.16 \pm 1.72$ |
| LwF | $19.61 \pm 0.05$ | $63.29 \pm 2.35$ | $9.70 \pm 0.23$ | $28.07 \pm 1.96$ | $7.65 \pm 1.31$ | $21.49 \pm 2.06$ |
| CPR | $21.38 \pm 0.19$ | $72.37 \pm 3.51$ | $10.31 \pm 0.24$ | $31.93 \pm 2.31$ | $9.85 \pm 0.23$ | $31.87 \pm 1.91$ |
| GPM | —— | $90.68 \pm 3.29$ | —— | $72.48 \pm 0.40$ | —— | $60.41 \pm 0.61$ |
| UCB | —— | $79.28 \pm 1.87$ | —— | $57.15 \pm 1.67$ | —— | $49.36 \pm 1.09$ |
| HAT | —— | $92.56 \pm 0.78$ | —— | $72.06 \pm 0.50$ | —— | $59.78 \pm 0.57$ |
| A-GEM | $22.67 \pm 0.57$ | $89.48 \pm 1.45$ | $9.30 \pm 0.32$ | $48.06 \pm 0.57$ | $7.76 \pm 0.12$ | $39.28 \pm 0.43$ |
| GSS | $49.73 \pm 4.78$ | $91.02 \pm 1.57$ | $13.60 \pm 2.98$ | $57.50 \pm 1.93$ | $11.22 \pm 3.17$ | $50.19 \pm 1.78$ |
| HAL | $41.79 \pm 4.46$ | $84.54 \pm 2.36$ | $9.05 \pm 2.76$ | $42.94 \pm 1.80$ | $4.46 \pm 1.72$ | $31.97 \pm 1.57$ |
| ER | $57.74 \pm 0.27$ | $93.61 \pm 0.27$ | $20.98 \pm 0.35$ | $73.37 \pm 0.43$ | $11.76 \pm 0.75$ | $61.58 \pm 0.31$ |
| ER+GMED | $57.89 \pm 0.38$ | $93.45 \pm 0.39$ | $21.07 \pm 0.43$ | $73.42 \pm 0.49$ | $11.85 \pm 0.81$ | $61.76 \pm 0.37$ |
| ER+DCMT | $62.07 \pm 0.43$ | $95.16 \pm 0.46$ | $23.54 \pm 0.51$ | $\mathbf{75.93 \pm 0.56}$ | $14.28 \pm 0.85$ | $64.37 \pm 0.53$ |
| ER+SCMT | $\mathbf{64.49 \pm 0.72}$ | $\mathbf{95.35 \pm 0.53}$ | $\mathbf{24.19 \pm 0.63}$ | $75.51 \pm 0.67$ | $\mathbf{14.95 \pm 0.93}$ | $\mathbf{64.91 \pm 0.67}$ |
| DER++ | $72.70 \pm 1.36$ | $93.88 \pm 0.50$ | $36.37 \pm 0.85$ | $75.64 \pm 0.60$ | $22.09 \pm 0.63$ | $61.26 \pm 0.57$ |
| DER++GMED | $72.82 \pm 1.79$ | $93.94 \pm 0.70$ | $36.25 \pm 0.69$ | $75.49 \pm 0.64$ | $22.21 \pm 0.81$ | $61.42 \pm 0.64$ |
| DER++noise | $72.41 \pm 1.83$ | $93.96 \pm 0.74$ | $36.02 \pm 0.91$ | $75.38 \pm 0.68$ | $22.32 \pm 0.87$ | $61.51 \pm 0.71$ |
| DER++DT(Ours) | $73.77 \pm 1.79$ | $94.68 \pm 0.82$ | $37.53 \pm 0.97$ | $77.21 \pm 0.79$ | $23.45 \pm 0.93$ | $63.03 \pm 0.75$ |
| DER++DCMT(Ours) | $74.86 \pm 1.53$ | $95.29 \pm 0.54$ | $\mathbf{38.68 \pm 0.81}$ | $78.56 \pm 0.82$ | $24.53 \pm 0.89$ | $64.08 \pm 0.78$ |
| DER++SCMT(Ours) | $\mathbf{75.12 \pm 1.62}$ | $\mathbf{95.67 \pm 0.67}$ | $38.56 \pm 0.93$ | $\mathbf{78.75 \pm 0.87}$ | $\mathbf{24.81 \pm 0.97}$ | $\mathbf{64.73 \pm 0.86}$ |

into 10 disjoint tasks, where each task has 10 classes. We also split CIFAR-100 into 10 disjoint tasks, where each task has 10 classes. The transformation time $T$ at each CL step is $T = 0.05$. Other hyperparameter settings follow (Buzzega et al., 2020). The memory buffer has a size of 500 data points by default. For the DCMT architecture, it is a four-block Resnet with a filter size of 8. For the SCMT architecture, the drift and diffusion networks are both four-block Resnet with a filter size of 6. DT (Section 3.2) uses the same architecture as DCMT. Thus, for DT, DCMT and SCMT, they have a much smaller number of parameters compared to the ResNet18 (He et al., 2016). It only accounts for about $0.4\%$ parameters of ResNet18, thus is negligible. To exclude the influence of the number of parameters in performance comparison, we reduce the number of parameters in our base model ResNet18 to offset the parameters in our transformation component. This ensures that all the compared models have the same number of parameters. All reported results in our experiments are the average accuracy and standard deviation with ten runs. The compared methods and our proposed methods are based on the public implementation [1].

**Result.** We compare the proposed methods to various CL baselines in Table 1. Due to space limitations, we put the results on *Tiny-ImageNet* in Appendix B.9 and backward transfer results in Appendix B.5. We can observe that our method outperforms those baselines. In particular, for class-IL, combining ER with DCMT or SCMT outperforms ER by $3.2\%$, $3.1\%$, and $6.7\%$, on MiniImageNet, CIFAR-100 and CIFAR10, respectively. For task-IL, integrating ER with DCMT or SCMT outperforms ER by $3.3\%$, $2.6\%$ and $1.7\%$ on MiniImageNet, CIFAR-100, and CIFAR10, respectively. Furthermore, for class-IL, combining DER++ with DCMT or SCMT, outperforms DER++ by $2.7\%$, $2.3\%$ and $2.4\%$ on MiniImageNet, CIFAR-100 and CIFAR10 respectively. For task-IL, integrating DER++ with DCMT or SCMT, outperforms DER++ by $3.5\%$, $3.1\%$, and $1.8\%$ on MiniImageNet, CIFAR-100 and CIFAR10 respectively. SCMT can further improve over DCMT due to the increased data diversity. GMED brings little or even worse performance, consistent with the observations of (Jin et al., 2021). We believe this is because, in task-aware CL, memory buffer data could be replayed many epochs, and GMED may edit the memory data too much so that they may significantly deviate from the original raw data. Adding random noise marginally helps in some cases, but some cases brings worse performance. We believe that adding noise only adds a simple transformation pattern, and still lacks diversity. When training for a long time, the network could also memorize the noise pattern. DT improves the performance compared to baselines since discrete step transformation by neural network can be viewed as discrete approximation of our continuous transformation. The performance of DT is upper bound by our continuous transformation. Also, DT needs to store intermediate transformations. In contrast, DCMT and SCMT do not need to store them. Our method outperforms baselines because the memory transformation function is highly expressive

[1] https://github.com/aimagelab/mammoth

and flexible and provides significantly more memory diversity. The transformed memory buffer is more difficult for the CL model to overfit.

**Ablation Study** (1) Due to space limitations, we put hyperparameter analysis, including $\lambda$, $T$, etc. in Appendix B.6. (2) Memory size 2000 in Appendix B.4. (3) Memory visualization in Appendix B.7.

## 4.2 TASK-FREE CL

**Baseline.** We performed experiments by comparing to the following task-free CL baselines, including ER (Chaudhry et al., 2019b), MIR (Aljundi et al., 2019a), *AGEM (Chaudhry et al., 2019a)*, GSS-Greedy (Aljundi et al., 2019c) and GMED (Jin et al., 2021) with single epoch training. Furthermore, following (Jin et al., 2021), we also compare data augmentation, such as random rotations, scaling, and horizontal flipping applied to ER method, named $ER_{aug}$. More descriptions of baselines are placed in Appendix B.2.

**Implementation details.** We use Resnet-18 as (Aljundi et al., 2019a). The transformation time $T$ at each CL step is $T = 0.03$. By default, following (Jin et al., 2021), we set the memory buffer size to be 500 for CIFAR-10, $10K$ for MiniImagenet, and $5K$ for CIFAR-100. Other hyperparameters are the same as (Aljundi et al., 2019a). All reported results are the average accuracy and standard deviation with ten runs.

Table 2: **Task-free CL** results on CIFAR10, CIFAR-100 and MiniImagenet, respectively.

| Algorithm | CIFAR10 | CIFAR-100 | MiniImagenet |
|---|---|---|---|
| fine-tuning | $18.9 \pm 0.1$ | $3.1 \pm 0.2$ | $2.9 \pm 0.5$ |
| A-GEM | $19.0 \pm 0.3$ | $2.4 \pm 0.2$ | $3.0 \pm 0.4$ |
| GSS-Greedy | $29.9 \pm 1.5$ | $19.5 \pm 1.3$ | $17.4 \pm 0.9$ |
| ER | $33.3 \pm 2.8$ | $20.1 \pm 1.2$ | $24.8 \pm 1.0$ |
| ER+DCMT | $36.5 \pm 2.6$ | $\mathbf{21.9 \pm 1.3}$ | $27.6 \pm 1.5$ |
| ER+SCMT | $\mathbf{37.3 \pm 3.0}$ | $21.6 \pm 1.5$ | $\mathbf{27.9 \pm 1.6}$ |
| MIR | $34.4 \pm 2.4$ | $20.0 \pm 1.7$ | $25.3 \pm 1.7$ |
| MIR+DCMT | $36.8 \pm 2.7$ | $21.3 \pm 1.8$ | $27.5 \pm 1.9$ |
| MIR+SCMT | $\mathbf{37.6 \pm 2.9}$ | $\mathbf{21.7 \pm 2.1}$ | $\mathbf{27.8 \pm 2.0}$ |
| GMED (ER) | $34.8 \pm 2.2$ | $20.9 \pm 1.6$ | $27.3 \pm 1.8$ |
| GMED+DCMT | $37.1 \pm 2.4$ | $21.6 \pm 1.8$ | $28.1 \pm 1.7$ |
| GMED+SCMT | $\mathbf{37.9 \pm 2.7}$ | $\mathbf{21.8 \pm 1.9}$ | $\mathbf{28.5 \pm 1.9}$ |
| $ER_{aug}$ | $46.3 \pm 2.7$ | $18.3 \pm 1.9$ | $30.8 \pm 2.2$ |
| $ER_{aug}$+DCMT | $\mathbf{48.2 \pm 2.9}$ | $20.4 \pm 2.1$ | $31.7 \pm 2.4$ |
| $ER_{aug}$+SCMT | $47.9 \pm 3.0$ | $\mathbf{20.6 \pm 2.5}$ | $\mathbf{32.1 \pm 2.3}$ |

**Result.** We compare to various CL baselines and combination with ER, MIR, and GMED in Table 2. We observe that our method outperforms these baselines. Particularly, combining ER with DCMT or SCMT outperforms ER by $4.0\%$, $3.1\%$, and $1.8\%$ on CIFAR10, MiniImageNet and CIFAR-100, respectively. Combining MIR with DCMT or SCMT outperforms MIR by $3.2\%$, $2.5\%$ and $1.7\%$ on CIFAR10, MiniImageNet and CIFAR-100. Combining GMED with DCMT or SCMT outperforms GMED by $3.1\%$,

Table 3: **Task-free CL** results on CIFAR-100 and MiniImagenet with **different memory size**

| Algorithm | CIFAR-100 | | MiniImagenet | |
|---|---|---|---|---|
| Memory size | 2000 | 3000 | 2000 | 5000 |
| ER | $11.2 \pm 1.0$ | $15.0 \pm 0.9$ | $11.0 \pm 0.3$ | $17.9 \pm 1.6$ |
| ER+ DCMT | $14.5 \pm 1.1$ | $17.1 \pm 1.1$ | $14.1 \pm 0.5$ | $21.0 \pm 1.7$ |
| ER+ SCMT | $\mathbf{14.9 \pm 1.3}$ | $\mathbf{17.6 \pm 1.4}$ | $\mathbf{14.5 \pm 0.6}$ | $\mathbf{21.3 \pm 1.9}$ |
| MIR | $11.6 \pm 0.8$ | $15.6 \pm 1.0$ | $11.2 \pm 0.4$ | $17.4 \pm 1.2$ |
| MIR+DCMT | $\mathbf{14.8 \pm 1.0}$ | $16.7 \pm 0.9$ | $14.6 \pm 0.8$ | $20.6 \pm 1.4$ |
| MIR+SCMT | $14.6 \pm 1.1$ | $\mathbf{17.0 \pm 1.2}$ | $\mathbf{15.0 \pm 1.1}$ | $\mathbf{21.0 \pm 1.6}$ |

$1.2\%$ and $0.9\%$ on CIFAR10, MiniImageNet and CIFAR-100. Our method outperforms baselines for reasons similar to task-aware CL.

**Ablation Study**: The smaller memory size of 2000 and 3000 for CIFAR100; memory sizes of 2000 and 5000 for Mini-imageNet are provided in Table 3. This further shows the significant improvement of DCMT and SCMT with more than $3\%$ improvements in many cases.

**Complexity analysis**. Our current implementation has comparable computation cost compared to GMED. We put our efficiency improvement techniques and detailed complexity analysis in Appendix B.8. We put the formal complexity analysis in Table 10 and run time versus performance evaluation in Table 11 in Appendix B.8.

## 5 CONCLUSION

This paper explores the memory overfitting issues and proposes a novel continuous memory transformer. We apply the proposed method to both task-aware and task-free CL. Compared to existing works, our proposed methods are very flexible and can make the memory data diverse and hard to overfit. Extensive experiments with both strong baselines of task-aware and task-free CL demonstrate the effectiveness of the proposed methods. Future work includes automatically learning the transformation time interval to obtain the optimal transformed memory data.

## REPRODUCIBILITY STATEMENT

We provided detailed implementation details and codebase we used to implement our methods.

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

## A    APPENDIX

We provide more experimental details and results in Section B and theoretical analysis in Section **??**.

## B    EXPERIMENTS

### B.1    MORE IMPLEMENTATION DETAILS

We use NVIDIA RTX A6000 GPU to do the experiments. For the hyperparameters values of DER++, ER and all the other baselines, we follow the implementation in (Buzzega et al., 2020).

### B.2    BASELINE DESCRIPTIONS

**Experience Replay (ER)**    (Chaudhry et al., 2019b) stores a small subset data from previous tasks with reservoir sampling (Chaudhry et al., 2019b). When training with new tasks, we randomly sample a subset of examples from the memory buffer to train with the received new mini-batch data together to mitigate forgetting.

**Maximally Interfering Retrieval (MIR)**    (Aljundi et al., 2019a), the goal of MIR is to select the examples that are most easily forgettable for replay. We follow similar setting in GMED (Jin et al., 2021) for a fair comparison. We evaluate the CL model forgetting with 25 memory examples for Mini-ImageNet dataset, and 50 memory examples for other datasets.

**Averaged Gradient Episodic Memory (AGEM)**    (Chaudhry et al., 2019a). At every CL training step, AGEM ensures that the average memory buffer data loss over the previous tasks does not increase. AGEM projects the gradient update direction to the closest gradient direction in $\mathcal{L}_2$ space that keeps the gradient angle less than 90 degree to ensure the memory data are less interfered with current data.

**Gradient-Based Sample Selection (GSS-Greedy)**    (Aljundi et al., 2019c) encourages storing diverse examples in the memory buffer. We use GSS-Greedy, which is efficient and performs the best in the variants proposed in (Aljundi et al., 2019c).

**Gradient based Memory Editing (GMED)**    (Jin et al., 2021). The goal of GMED is to edit the memory buffer data with gradient information so that they are harder to be memorized, which shares the similar goal as MIR(Aljundi et al., 2019a).

**Dark Experience Replay (DER)**    (Buzzega et al., 2020) is a memory replay-based methods that combines memory replay with knowledge distillation to mitigate forgetting. DER++ is one of the state-of-art methods in CL.

**Hindsight Anchor Learning (HAL)**    (Chaudhry et al., 2021) regularizes the training objective with one data point per class per task, named anchors by maximizing its estimated forgetting. Keeping the model prediction fixed on those anchor points preserves the performance of previous tasks.

**Task-free CL additional baselines**    Following (Aljundi et al., 2019a), we also additionally compare: (1) *iid online*: which trains the model with a single-pass through the iid sampled data on the same set of samples; (2) *iid offline*: which trains the model with multiple epochs through the iid sampled data. We train the model with 5 epochs for this baseline and the performance serves as upper-bound. Table 4 shows the results of these two baselines.

Table 4: Results for Task-free CL *iid online and iid offline* training on CIFAR10, CIFAR-100 and MiniImagenet, respectively.

| Algorithm | CIFAR10 | CIFAR-100 | MiniImagenet |
|---|---|---|---|
| iid online | $60.3 \pm 1.4$ | $18.7 \pm 1.2$ | $17.7 \pm 1.5$ |
| iid offline | $78.7 \pm 1.1$ | $42.0 \pm 0.9$ | $39.8 \pm 1.4$ |

### B.3 BI-LEVEL OPTIMIZATION

$$\boldsymbol{x}^m(T) = \boldsymbol{x}^m(0) + \int_0^T g(\boldsymbol{x}^m(t), t, \boldsymbol{\phi})dt \tag{11}$$

We first define adjoint-state at time $t$ as $\boldsymbol{a}(t) = \frac{d\mathcal{L}(\boldsymbol{x}^m(T), y)}{d\boldsymbol{x}^m(t)} = \frac{d\mathcal{L}}{d\boldsymbol{x}^m(t)}$, here we abbreviate the notation $\mathcal{L}(\boldsymbol{x}^m(T), y)$ as $\mathcal{L}$, which follows the following differential equation:

$$\frac{d\boldsymbol{a}(t)}{dt} = -\boldsymbol{a}(t)^T \frac{\partial g(\boldsymbol{x}^m(t), t, \boldsymbol{\phi})}{\boldsymbol{x}^m}, \quad \boldsymbol{a}_T = \frac{d\mathcal{L}}{d\boldsymbol{x}^m(T)} \tag{12}$$

The adjoint-state equation Eq. (12) can be viewed as the continuous version of backpropagation (for discrete number of layers). Therefore, we can obtain $\boldsymbol{a}(0) = \frac{d\mathcal{L}}{d\boldsymbol{x}^m(0)}$ by the reverse time ODE in the time interval $[T, 0]$ with initial state, i.e., $\boldsymbol{a}_T = \frac{d\mathcal{L}}{d\boldsymbol{x}^m(T)}$. We can compute the gradient with respect to the memory transformer parameters $\boldsymbol{\phi}$ as the following equation:

$$\frac{\partial \mathcal{L}}{\partial \boldsymbol{\phi}} = -\int_T^0 \boldsymbol{a}(t)^T \frac{\partial g(\boldsymbol{x}^m(t), t, \boldsymbol{\phi})}{\partial \boldsymbol{\phi}} \tag{13}$$

The Eq. (11) serves as for transforming the memory data. Eq.(12) serves as for calculating the adjoint state $\boldsymbol{a}(t)$. Eq. (13) calculates the gradient with respect to the memory transformer $\boldsymbol{\phi}$. Those three integration can be jointly solved together in a single pass by concatenating the transformed memory data, the adjoint state and derivatives with respect to the parameters. The entire algorithm for calculating the derivative with respect to the memory transformer parameters is shown in Algorithm 2 in the following.

---

**Algorithm 2** Reverse-mode derivative for calculating the gradient w.r.t $\boldsymbol{\phi}$ of DCMT.

---

1: **REQUIRE:** memory transformer parameters $\boldsymbol{\phi}$, transformation time interval $[0, T]$, final transformation state at time $T$, i.e., $\boldsymbol{x}^m(T)$, loss gradient $\frac{\partial \mathcal{L}}{\partial \boldsymbol{x}^m(T)}$.
2: $s_0 = [\boldsymbol{x}^m(T), \frac{\partial \mathcal{L}}{\partial \boldsymbol{x}^m(T)}, \boldsymbol{0}_{|\boldsymbol{\phi}|}]$;   calculate adjoint state $\boldsymbol{a}_t = \frac{d\mathcal{L}}{d\boldsymbol{x}^m(t)}$
3: $\boldsymbol{def}$ aug-dynamics($[\boldsymbol{x}^m(t), \boldsymbol{a}_t, \cdot]$):
4:    $\boldsymbol{return}[g(\boldsymbol{x}^m(t), t, \boldsymbol{\phi}), -\boldsymbol{a}_t^T \frac{\partial g}{\partial \boldsymbol{x}^m}, -\boldsymbol{a}_t^T \frac{\partial g}{\partial \boldsymbol{\phi}}]$
5: $[\boldsymbol{x}^m(0), \frac{\partial \mathcal{L}}{\partial \boldsymbol{x}^m(0)}, \frac{\partial \mathcal{L}}{\partial \boldsymbol{\phi}}] = \text{ODEsolver}(s_0, aug - dynamics, T, 0)$
6: **Return** $\frac{\partial \mathcal{L}}{\partial \boldsymbol{x}^m(0)}, \frac{\partial \mathcal{L}}{\partial \boldsymbol{\phi}}$

---

### B.4 TASK-AWARE (CLASS-IL AND TASK-IL) MEMORY SIZE 2000

Table 5 shows the results for task-IL and class-IL on CIFAR-100 and MiniImagenet, respectively with memory size 2000. Our methods still significantly outperform strong baselines by a large margin.

Table 5: **Task-IL and class-IL** results on CIFAR-100 and MiniImagenet, respectively with **memory size 2000**

| Algorithm Method | CIFAR-100 | | MiniImagenet | |
|---|---|---|---|---|
| | Class-IL | Task-IL | Class-IL | Task-IL |
| A-GEM | $9.36 \pm 0.39$ | $49.62 \pm 1.08$ | $8.10 \pm 0.28$ | $38.75 \pm 0.89$ |
| GSS | $16.31 \pm 3.76$ | $65.52 \pm 1.86$ | $14.01 \pm 3.64$ | $55.39 \pm 2.03$ |
| HAL | $15.61 \pm 3.94$ | $48.28 \pm 2.04$ | $7.88 \pm 2.78$ | $32.91 \pm 1.75$ |
| ER | $36.87 \pm 0.64$ | $82.07 \pm 0.42$ | $22.78 \pm 0.57$ | $72.25 \pm 0.32$ |
| ER+ GMED | $36.35 \pm 0.77$ | $81.38 \pm 0.45$ | $22.86 \pm 0.65$ | $72.16 \pm 0.39$ |
| ER+ DCMT | $38.90 \pm 0.71$ | $83.45 \pm 0.51$ | $24.94 \pm 0.78$ | **74.57±0.46** |
| ER+ SCMT | **39.28±0.89** | **83.79 ±0.68** | **25.37±0.87** | $74.31 \pm 0.57$ |
| DER++ | $50.72 \pm 0.71$ | $82.43 \pm 0.38$ | $37.82 \pm 0.79$ | $72.02 \pm 0.31$ |
| DER++GMED | $51.03 \pm 0.68$ | $82.06 \pm 0.41$ | $37.98 \pm 0.85$ | $71.68 \pm 0.41$ |
| DER++DCMT | $52.93 \pm 0.94$ | $83.81 \pm 0.52$ | **40.16±0.84** | $74.06 \pm 0.49$ |
| DER++SCMT | **52.97±0.73** | **84.26± 0.69** | $39.91 \pm 0.89$ | **74.29±0.65** |

## B.5 BACKWARD TRANSFER (BWT)

Table 6 shows the backward transfer results with memory size 500 on various datasets. BWT measures the forgetting of CL model. Note that if one method restrains learning the current task would preserve the past knowledge with high BWT but achieves overall low accuracy. This would make the current task not learned well. The results indicate the significant improvement of the proposed methods (DCMT and SCMT) for mitigating forgetting compared to baseline ER and DER++. The baseline HAL achieves higher BWT in some cases because HAL does not learn the new task well and achieves overall much lower accuracy as shown in Table 1 (main text).

Table 6: **Backward Transfer** of various methods with **memory size 500**.

| Method | CIFAR10 | | CIFAR100 | | MiniImageNet | |
|---|---|---|---|---|---|---|
| | Class-IL | Task-IL | Class-IL | Task-IL | Class-IL | Task-IL |
| finetuning | $-96.39 \pm 0.12$ | $-46.24 \pm 2.12$ | $-89.68 \pm 0.96$ | $-62.46 \pm 0.78$ | $-82.48 \pm 1.17$ | $-61.5 \pm 0.82$ |
| AGEM | $-94.01 \pm 1.16$ | $-14.26 \pm 1.18$ | $-88.5 \pm 1.56$ | $-45.43 \pm 2.32$ | $-79.86 \pm 1.43$ | $-44.84 \pm 2.81$ |
| GSS | $-62.88 \pm 2.67$ | $-7.73 \pm 3.99$ | $-82.17 \pm 4.16$ | $-33.98 \pm 1.54$ | $-77.21 \pm 3.97$ | $-34.12 \pm 1.85$ |
| HAL | $-62.21 \pm 4.34$ | $-5.41 \pm 1.10$ | $-49.29 \pm 2.82$ | $-13.60 \pm 1.04$ | $-40.55 \pm 2.69$ | $-10.50 \pm 1.70$ |
| ER | $-45.35 \pm 0.07$ | $-3.54 \pm 0.35$ | $-74.84 \pm 1.38$ | $-16.81 \pm 0.97$ | $-78.21 \pm 1.57$ | $-21.96 \pm 0.32$ |
| ER+GMED | $-45.42 \pm 0.31$ | $-3.49 \pm 0.33$ | $-74.60 \pm 1.26$ | $-16.97 \pm 0.91$ | $-77.76 \pm 1.38$ | $-22.10 \pm 0.27$ |
| ER+DCMT | $-43.91 \pm 0.09$ | $-3.15 \pm 0.46$ | $-73.31 \pm 1.51$ | $-15.23 \pm 1.08$ | **-76.74 ± 1.72** | $-20.57 \pm 0.43$ |
| ER+SCMT | **-43.65 ± 0.11** | **-3.12 ± 0.53** | **-73.19 ±1.67** | **-15.02 ± 1.21** | $-76.92 \pm 1.89$ | **-20.43 ± 0.58** |
| DER++ | $-22.38 \pm 4.41$ | $-4.66 \pm 1.15$ | $-53.89 \pm 1.85$ | $-14.72 \pm 0.96$ | $-61.85 \pm 2.73$ | $-22.93 \pm 0.91$ |
| DER++GMED | $-22.53 \pm 4.21$ | $-4.57 \pm 1.15$ | $-53.67 \pm 1.71$ | $-14.86 \pm 0.94$ | $-61.72 \pm 2.61$ | $-22.71 \pm 0.87$ |
| DER++DCMT | $-20.61 \pm 4.58$ | $-4.21 \pm 1.26$ | **-52.03 ± 1.93** | $-13.92 \pm 0.97$ | $-59.49 \pm 2.89$ | $-20.51 \pm 0.98$ |
| DER++SCMT | **-20.27 ± 4.67** | **-4.08 ± 1.37** | $-52.31 \pm 2.06$ | **-13.61 ± 1.09** | **-59.03 ± 2.97** | **-20.09 ± 1.07** |

## B.6 HYPERPARAMETER ANALYSIS

Table 7 shows the effect of integration (transformation) time $T$ on the performance. We can observe that the performance becomes better with a longer integration time $T$. This is because with a longer integration time $T$, we can generate more diverse transformed memory data so that the proposed methods generalize better to previous tasks. If the $T$ is too large ($T = 2.0$), the transformed data would significantly deviate from the original data distribution, thus the performance could become worse. To maintain the transformed memory data distribution not deviate from the original data distribution too much and reduce computation cost, we use a moderate time $T$. Table 8 shows the effect of $\lambda$ on the model performance. The model achieves best performance at $\lambda = 3.0$. Table 9 shows the effect of the number of editing steps $N$ on the performance for *combining GMED (Jin et al., 2021) with DER++*. We can observe that with more editing steps, the performance of GMED drops significantly because the memory data becomes much harder but lacks data diversity, thus generalizing worse with more editing steps.

Table 7: Effect of **integration (transformation) time** $T$ for combining DER++ with DCMT on CIFAR10, CIFAR-100 and MiniImagenet, respectively.

| | Integration time | $T = 0.03$ | $T = 0.05$ | $T = 2.0$ |
|---|---|---|---|---|
| Task-IL | **CIFAR10** | $94.78 \pm 0.42$ | $95.29 \pm 0.54$ | $94.42 \pm 0.64$ |
| | **MiniImagenet** | $63.42 \pm 0.67$ | $64.08 \pm 0.78$ | $63.26 \pm 0.90$ |
| Class-IL | **CIFAR10** | $74.51 \pm 1.37$ | $74.86 \pm 1.53$ | $74.17 \pm 1.49$ |
| | **MiniImagenet** | $24.16 \pm 0.72$ | $24.53 \pm 0.89$ | $23.93 \pm 0.96$ |

Table 8: Effect of **regularization weight** $\lambda$ for combining DER++ with DCMT on CIFAR10, CIFAR-100 and MiniImagenet, respectively.

| regularization weight | $\lambda = 0.0$ | $\lambda = 1.0$ | $\lambda = 3.0$ | $\lambda = 5.0$ |
|---|---|---|---|---|
| **CIFAR10** | $94.62 \pm 0.49$ | $94.98 \pm 0.54$ | $95.29 \pm 0.54$ | $94.85 \pm 0.57$ |
| **MiniImagenet** | $63.71 \pm 0.72$ | $64.37 \pm 0.53$ | $64.08 \pm 0.78$ | $64.11 \pm 0.73$ |

Table 9: **Effect of the number of editing steps** $N$ **for DER++GMED (Jin et al., 2021)** on CIFAR10, CIFAR-100 and MiniImagenet, respectively. We can observe that with more editing steps, the performance of GMED drops significantly because the memory data becomes much harder but lacks data diversity, thus generalizing worse with more editing steps.

| | number of editing steps | $N = 1$ | $N = 3$ | $N = 5$ | $N = 10$ |
|---|---|---|---|---|---|
| Task-IL | **CIFAR10** | $93.94 \pm 0.70$ | $93.47 \pm 0.67$ | $90.92 \pm 0.61$ | $88.68 \pm 0.79$ |
| | **MiniImagenet** | $61.42 \pm 0.64$ | $58.36 \pm 0.61$ | $52.19 \pm 0.57$ | $46.52 \pm 0.42$ |
| Class-IL | **CIFAR10** | $72.82 \pm 1.79$ | $70.59 \pm 1.62$ | $60.96 \pm 1.51$ | $59.40 \pm 0.57$ |
| | **MiniImagenet** | $22.21 \pm 0.81$ | $19.83 \pm 0.71$ | $15.96 \pm 0.62$ | $14.54 \pm 0.53$ |

## B.7 MEMORY TRANSFORMATION VISUALIZATION

Figure 3 and 4 show the transformation results at different time $t$. With longer transformation time, the transformed data becomes diverse when viewing all the data in the transformation interval $[0, T]$. SCMT brings more diversity than DCMT in terms of appearance. This suggests that with gradual and continuous transformation of memory data can significantly improve the memory data diversity.

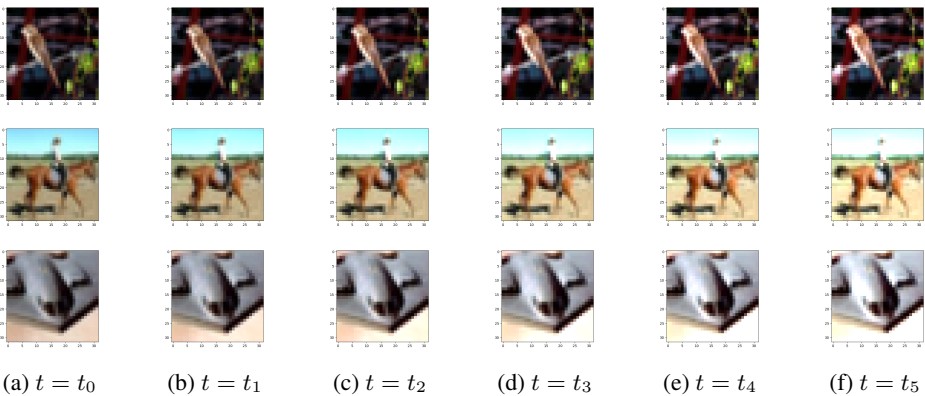

(a) $t = t_0$  (b) $t = t_1$  (c) $t = t_2$  (d) $t = t_3$  (e) $t = t_4$  (f) $t = t_5$

Figure 3: Gradual memory transformation by DCMT on CIFAR10.

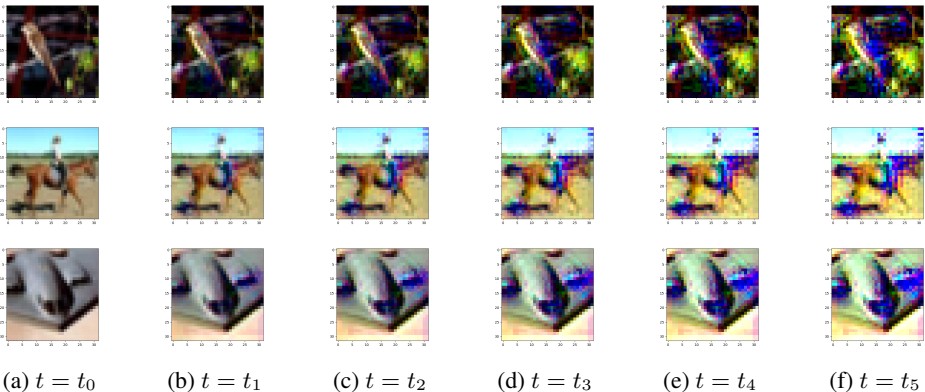

(a) $t = t_0$     (b) $t = t_1$     (c) $t = t_2$     (d) $t = t_3$     (e) $t = t_4$     (f) $t = t_5$

Figure 4: Gradual memory transformation by SCMT on CIFAR10.

### B.8   COMPLEXITY ANALYSIS AND EFFICIENCY EVALUATION

According to (Xu et al., 2022), the computation cost of backward propagation is two times of forward propagation.

- DER++: it needs 3 forward and 1 backward pass to replay the memory data, which equivalently requires 3 + 1*2 =5 forward computation.

- DER++GMED: at each CL step, GMED additionally needs 3 forwards and 1 backward passes (for two mini-batch data) to edit the memory examples for each editing step. So, the complexity is 6+2*2= 10 forward propagation calculations.

- DER++DCMT, since we uniformly sample the number $n$ from [0, 5], the amortized cost across different CL steps is to use 3 discrete time points at each CL step in the interval $[0, T]$. First, the time complexity of forward ODE calculation at those time points is 3 forward propagation. Then, we need to calculate the gradients for model parameters. The time complexity 3 backward gradient calculation at those time points. Thus, the computation cost of DCMT is equivalent to 6+3*2 forward propagation. We can further reduce the computation cost of DCMT by backward propagation into the memory transformer for every $S$ CL step instead of backpropagating into the memory transformer for every CL step. For example, suppose we have 3 CL steps, i.e., 1, 2, 3; we backpropagate into the memory transformer at step 1, but use the same memory transformer in the step 2 and 3 to transform the memory data. This method further reduces the cost of DCMT and SCMT into $6 + \frac{6}{3} = 8$. Additionally, the integration and backward computation time for the memory transformer in the interval $[0, T]$ is empirically equivalent to 1.6 forward computation of the ResNet backbone since the memory transformer network is very small. Thus, the total computation cost is 9.6 forward computation.

- DER++SCMT, the complexity is similar, but it requires evaluating both the drift and diffusion terms with the corresponding gradient. Thus, the computation cost doubles the cost of DCMT, and therefore it is equivalent to $6 + \frac{6}{3} = 8$ forward propagation. Additionally, the integration and backward computation time for the memory transformer in the interval $[0, T]$ is empirically equivalent to 4.2 forward computation of the ResNet backbone since the memory transformer network is very small. Thus, the total computation cost is 12.2 forward computation.

We summarize the computation complexity of the above methods in table 10. We set DER++ as the baseline with a running time unit of 1; we compare the computation complexity of all the other methods with respect to DER++.

Table 11 shows the running time evaluation of different methods on CIFAR-100 for one epoch training.

Table 10: Formal analysis of computation complexity of different methods.

| Method | running time (units) |
|--------|---------------------|
| DER++ | 1.0 |
| DER++GMED | 2.0 |
| DER++DCMT | 1.9 |
| DER++SCMT | 2.4 |

Table 11: Performance verses efficiency evaluation (wall clock time) on CIFAR-100 for one epoch training. '——' indicates not applicable.

| Method | Class-IL | Task-IL | running time (wall clock) |
|--------|----------|---------|---------------------------|
| GSS | $13.6 \pm 2.98$ | $57.50 \pm 1.93$ | 50.6 |
| HAL | $9.05 \pm 2.76$ | $42.94 \pm 1.80$ | 27.9 |
| UCB | —— | $57.15 \pm 1.67$ | 83.6 |
| DER++ | $36.37 \pm 0.85$ | $75.64 \pm 0.60$ | 25.8 |
| DER++GMED | $36.25 \pm 0.69$ | $75.49 \pm 0.64$ | 41.7 |
| DER++DCMT | $38.68 \pm 0.81$ | $78.56 \pm 0.82$ | 38.9 |
| DER++SCMT | $38.56 \pm 0.93$ | $78.75 \pm 0.87$ | 59.1 |

## B.9 RESULTS ON TINY-IMAGENET

Following (Buzzega et al., 2020), we split Tiny-ImageNet with 200 classes into 10 tasks. Each task has 20 classes. The results are shown in Table 12.

Table 12: **Task-IL and class-IL** results on Tiny-ImageNet, respectively with **memory size 500**

| Algorithm Method | Tiny-ImageNet | |
|--------|----------|---------|
| | Class-IL | Task-IL |
| oEWC | $7.58 \pm 0.10$ | $19.20 \pm 0.31$ |
| SI | $6.58 \pm 0.31$ | $36.32 \pm 0.13$ |
| LwF | $8.46 \pm 0.22$ | $15.85 \pm 0.58$ |
| CPR | $8.91 \pm 0.15$ | $20.71 \pm 0.35$ |
| UCB | —— | $46.89 \pm 0.42$ |
| A-GEM | $8.06 \pm 0.04$ | $25.33 \pm 0.49$ |
| ER | $9.99 \pm 0.29$ | $48.64 \pm 0.46$ |
| DER++ | $19.38 \pm 1.41$ | $51.91 \pm 0.68$ |
| DER++GMED | $18.93 \pm 1.56$ | $51.53 \pm 0.76$ |
| DER++DCMT | $21.06 \pm 1.26$ | $\mathbf{54.02 \pm 0.57}$ |
| DER++SCMT | $\mathbf{21.27 \pm 1.42}$ | $53.67 \pm 0.73$ |

## B.10 T-SNE VISUALIZATION

More T-SNE visualization of ER is shown in Figure 5, GMED is shown in Figure 6, our method is shown in Figure 7.

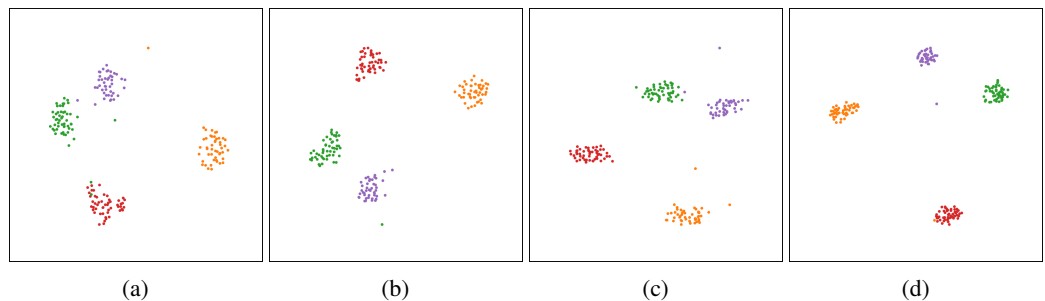

(a)      (b)      (c)      (d)

Figure 5: **Experience Replay (ER)** T-SNE visualization on CIFAR10. We use features extracted from the last layer output of ResNet18 as the input to T-SNE. We use four classes of memory data to illustrate the difference. T-SNE embeds each data point, and each color denotes one class of memory buffer data. Each figure is a plot at different times.

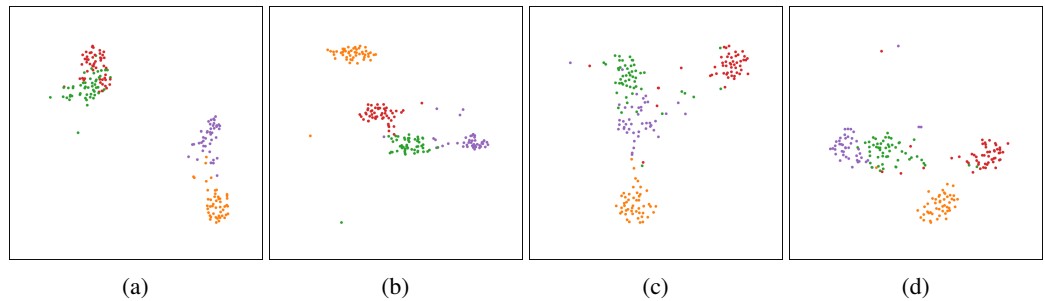

(a)      (b)      (c)      (d)

Figure 6: **GMED** T-SNE visualization on CIFAR10. We use features extracted from the last layer output of ResNet18 as the input to T-SNE. We use four classes of memory data to illustrate the difference. T-SNE embeds each data point, and each color denotes one class of memory buffer data. Each figure is a plot at different times.

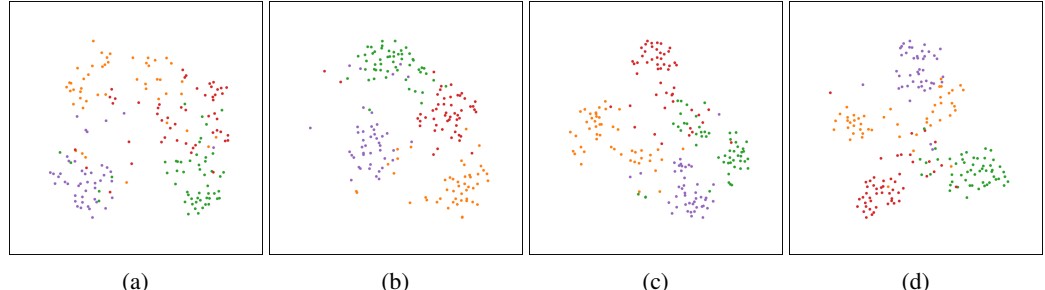

(a)      (b)      (c)      (d)

Figure 7: **DCMT (our method)** on CIFAR10 with T-SNE visualization. We use features extracted from the last layer output of ResNet18 as the input to T-SNE. We use four classes of memory data to illustrate the difference. T-SNE embeds each data point, and each color denotes one class of memory buffer data. Each figure is a plot at different times.

## B.11 Math Background

**Definition 1 (Stratonovich stochastic integral)** *Suppose $\{\mathcal{F}_{s,t}\}_{s<t;s,t\in[0,T]}$ be a two-sided filtration, where $\mathcal{F}_{s,t}$ is the $\sigma$-algebra generated by $\{W_u - W_v : s \leq u \leq v \leq t\}$ for $s, t \in [0, T]$ such that $s \leq t$. For a continuous semi-martingale $\{X_t\}_{t\in[0,T]}$ adapted to the forward filtration, $\{\mathcal{F}_{0,t}\}_{t\in[0,T]}$, Stratonovich stochastic integral is*

$$\int_0^T X_t \circ dW_t = \lim_{|\prod|\to 0} \sum_{k=1}^N (\frac{X_{t_k} + X_{t_{k-1}}}{2})(W_{t_k} - W_{t_{k-1}}) \tag{14}$$

*where $\prod = 0 = t_0 < \cdots < t_N = T$ is a partition of the interval $[0, T]$, and $|\prod| = \max_k t_k - t_{k-1}$ denotes the size of largest segment of the partition, and the limit is to be interpreted in the $L_2$ space.*

### B.12 TRAINING ALGORITHMS FOR DCMT AND SCMT

We present the Runge-Kutta Method in Algorithm 3 for integrating ODE to obtain its solution. We present the adjoint method for calculating the gradient with respect to the parameters of DCMT $\phi$ in Algorithm 2. We then present the algorithm for calculating the forward pass of SCMT (Eq. 7) in Algorithm 4. We then present the reversible Heun algorithm for calculating the gradients with respect to the parameters of SCMT in Algorithm 5.

---

**Algorithm 3** Runge-Kutta Method for integrating ODE.

---

1: **REQUIRE:** $\frac{d\boldsymbol{x}^m(t)}{dt} = g(\boldsymbol{x}^m(t), t, \boldsymbol{\phi}), \quad \boldsymbol{x}^m(0) = \boldsymbol{x}^m$;
2: dividing the time interval $[0, T]$ into sub-intervals, $[t_0, t_1], \cdots, \cdots, [t_n, t_{n+1}], \cdots, [t_{N-1}, t_N]$; where $|t_{n+1} - t_n| = h$; $h$ is the sep size.
3: **for** $n = 1$ to $N$ **do**
4:     $K_1 = g(\boldsymbol{x}_n^m, t_n)h$
5:     $K_2 = g(\boldsymbol{x}_n^m + \frac{1}{2}K_1, t_n + \frac{1}{2}h)h$
6:     $K_3 = g(\boldsymbol{x}_n^m + \frac{1}{2}K_2, t_n + \frac{1}{2}h)h$
7:     $K_4 = g(\boldsymbol{x}_n^m + K_3, t_n + h)h$
8:     $\boldsymbol{x}_{n+1}^m = \boldsymbol{x}_n^m + \frac{1}{6}(K_1 + 2K_2 + 2K_3 + K_4)$
9: **end for**
10: **Return** $\boldsymbol{x}_N^m$

---

**Algorithm 4** SCMT (SDE) solver (forward pass).

---

1: **REQUIRE:** $t_n, \boldsymbol{x}_n^m, \widehat{\boldsymbol{x}_n^m}, \mu_n, \sigma_n, \Delta t$, Brownian motion $W$.
2: $t_{n+1} = t_n + \Delta t$
3: $\Delta W_n = W_{t_{n+1}} - W_{t_n}$
4: $\widehat{\boldsymbol{x}_{n+1}^m} = 2\boldsymbol{x}_n^m - \widehat{\boldsymbol{x}_n^m} + \mu_n \Delta t + \sigma_n \Delta W_n$
5: $\mu_{n+1} = \mu(t_{n+1}, \widehat{\boldsymbol{x}_{n+1}^m})$
6: $\sigma_{n+1} = \sigma(t_{n+1}, \widehat{\boldsymbol{x}_{n+1}^m})$
7: $\boldsymbol{x}_{n+1}^m = \boldsymbol{x}_n^m + \frac{1}{2}(\mu_n + \mu_{n+1})\Delta t + \frac{1}{2}(\sigma_n + \sigma_{n+1})\Delta W_n$
8: **Return** $t_{n+1}, \boldsymbol{x}_{n+1}^m, \widehat{\boldsymbol{x}_{n+1}^m}, \mu_{n+1}, \sigma_{n+1}$

---

#### B.12.1 REVERSIBLE HEUN (KIDGER ET AL., 2021) FOR SOLVING THE SCMT GRADIENTS

Suppose we have the loss $\mathcal{L}$ on the terminal random variable $X_T$, then the adjoint process $A_t = \frac{d\mathcal{L}(X_T)}{dX_t} \in \mathcal{R}^d$ is the solution to the following SDE:

$$dA_t^i = -A_t^j \frac{\partial \mu^j}{\partial X^i}(t, X_t)dt - A_t^j \frac{\partial \sigma^{j,k}}{\partial X^i}(t, X_t) \circ dW_t^k \tag{15}$$

where $A_0 = \frac{d\mathcal{L}(X_T)}{dX_0}$ is the obtained backpropagated gradient. The gradients with respect to the parameters of the drift and diffusion networks can be obtained by viewing them as an additional part of the state whose dynamics has zero drift and diffusion (Li et al., 2020). Furthermore, the adjoint method of reversible Heun (Kidger et al., 2021) utilizes the reversibility of a differential equation: intermediate computations such as $X_t$ for $t < T$ are restored from $X_T$, so that they do not need to be held in memory.

---

**Algorithm 5** Reversible Heun method of Backward computation to calculate the gradients w.r.t the previous state of SCMT.

---

1: **REQUIRE:** $t_{n+1}, \boldsymbol{x}_{n+1}^m, \widehat{\boldsymbol{x}_{n+1}^m}, \mu_{n+1}, \sigma_{n+1}, \Delta t,$ Brownian motion $W$, $\frac{\partial \mathcal{L}(\boldsymbol{x}_T^m)}{\partial \boldsymbol{x}_{n+1}^m}, \frac{\partial \mathcal{L}(\boldsymbol{x}_T^m)}{\partial \widehat{\boldsymbol{x}_{n+1}^m}}, \frac{\partial \mathcal{L}(\boldsymbol{x}_T^m)}{\partial \mu_{n+1}}, \frac{\partial \mathcal{L}(\boldsymbol{x}_T^m)}{\partial \sigma_{n+1}}.$

2: $t_n = t_{n+1} - \Delta t$
3: $\Delta W_n = W_{t_{n+1}} - W_{t_n}$
4: $\widehat{\boldsymbol{x}_n^m} = 2\boldsymbol{x}_{n+1}^m - \widehat{\boldsymbol{x}_{n+1}^m} - \mu_{n+1}\Delta t - \sigma_{n+1}\Delta W_n$
5: $\mu_n = \mu(t_n, \widehat{\boldsymbol{x}_n^m})$
6: $\sigma_n = \sigma(t_n, \widehat{\boldsymbol{x}_n^m})$
7: $\boldsymbol{x}_n^m = \boldsymbol{x}_{n+1}^m - \frac{1}{2}(\mu_n + \mu_{n+1})\Delta t - \frac{1}{2}(\sigma_n + \sigma_{n+1})\Delta W_n$
8: $\boldsymbol{x}_{n+1}^m, \widehat{\boldsymbol{x}_{n+1}^m}, \mu_{n+1}, \sigma_{n+1} = Forward(t_n, \boldsymbol{x}_n^m, \widehat{\boldsymbol{x}_n^m}, \mu_n, \sigma_n, \Delta t, W)$
9: **Local Backpropagation**
10: $\frac{\partial \mathcal{L}(\boldsymbol{x}_T^m)}{\partial (\boldsymbol{x}_n^m, \widehat{\boldsymbol{x}_n^m}, \mu_n, \sigma_n)} = \frac{\partial \mathcal{L}(\boldsymbol{x}_T^m)}{\partial (\boldsymbol{x}_{n+1}^m, \widehat{\boldsymbol{x}_{n+1}^m}, \mu_{n+1}, \sigma_{n+1})} \frac{\partial (\boldsymbol{x}_{n+1}^m, \widehat{\boldsymbol{x}_{n+1}^m}, \mu_{n+1}, \sigma_{n+1})}{\partial (\boldsymbol{x}_n^m, \widehat{\boldsymbol{x}_n^m}, \mu_n, \sigma_n)}$
11: **Return** $t_n, \boldsymbol{x}_n^m, \widehat{\boldsymbol{x}_n^m}, \mu_n, \sigma_n, \frac{\partial \mathcal{L}(\boldsymbol{x}_T^m)}{\partial \boldsymbol{x}_n^m}, \frac{\partial \mathcal{L}(\boldsymbol{x}_T^m)}{\partial \widehat{\boldsymbol{x}_n^m}}, \frac{\partial \mathcal{L}(\boldsymbol{x}_T^m)}{\partial \mu_n}, \frac{\partial \mathcal{L}(\boldsymbol{x}_T^m)}{\partial \sigma_n}$

---

### B.13 WHY THE ENTIRE TRANSFORMATION IS INVERTIBLE AND THE VELOCITY TERM DOES NOT NEED TO BE INVERTIBLE

The forward integration is as the following:

$$\boldsymbol{x}^m(T) = \boldsymbol{x}^m(0) + \int_0^T g(\boldsymbol{x}^m(t), t, \boldsymbol{\phi})dt \tag{16}$$

We then multiple $-1$ for both sides of the above equations, obtain the following equation:

$$-\boldsymbol{x}^m(T) = -\boldsymbol{x}^m(0) - \int_0^T g(\boldsymbol{x}^m(t), t, \boldsymbol{\phi})dt \tag{17}$$

We then rearrange the above equation as following:

$$\boldsymbol{x}^m(0) = \boldsymbol{x}^m(T) - \int_0^T g(\boldsymbol{x}^m(t), t, \boldsymbol{\phi})dt \tag{18}$$

Then due to the following property of integration

$$-\int_0^T g(\boldsymbol{x}^m(t), t, \boldsymbol{\phi})dt = \int_T^0 g(\boldsymbol{x}^m(t), t, \boldsymbol{\phi})dt \tag{19}$$

We can obtain the following reverse integration:

$$\boldsymbol{x}^m(0) = \boldsymbol{x}^m(T) + \int_T^0 g(\boldsymbol{x}^m(t), t, \boldsymbol{\phi})dt \tag{20}$$

While we can observe that in the above derivation, there is no restriction that $g$ should be invertible.

## C PRELIMINARY AND RELATED WORK

### C.1 TASK-AWARE CL

**Problem setup.** Task-aware CL focuses on the case where there are explicit task definitions during CL. Task/domain/class-incremental learning (van de Ven & Tolias, 2019) are the three most representative CL scenarios. We consider the problem of learning a sequence of tasks denoted as $\mathcal{D}^{tr} = \{\mathcal{D}_1^{tr}, \mathcal{D}_2^{tr}, \cdots, \mathcal{D}_N^{tr}\}$, where $N$ is the number of training tasks. The $k$-th task training data

$\mathcal{D}_k^{tr}$ consists of a set of triplets $\{(\boldsymbol{x}_i^k, y_i^k, \mathcal{T}_k)_{i=1}^{n_k}\}$, where $\boldsymbol{x}_i^k$ is the $i$-th data example in the task, $y_i^k$ is the corresponding data label, and $\mathcal{T}_k$ is the task identifier. The goal is to learn a model $f_{\boldsymbol{\theta}}$ on the training task sequence $\mathcal{D}^{tr}$ so that it performs well on the test set of all the learned tasks $\mathcal{D}^{te} = \{\mathcal{D}_1^{te}, \mathcal{D}_2^{te}, \cdots, \mathcal{D}_N^{te}\}$ without forgetting previously learned knowledge.

**Existing work.** The proposed approaches for task-aware CL can be categorized into: 1) maintaining a memory buffer that stores previous examples for future replay (Lopez-Paz & Ranzato, 2017; Shin et al., 2017; Chaudhry et al., 2019a; Riemer et al., 2019; Chaudhry et al., 2019b; Aljundi et al., 2019a; PourKeshavarzi et al., 2022; Arani et al., 2022); 2) using dynamic network architectures (Rusu et al., 2016; Fernando et al., 2017; Yoon et al., 2018; Qin et al., 2021; Miao et al., 2022) and remembering past knowledge by dynamically updated architectures; 3) enforcing regularization to slow down forgetting (Kirkpatrick et al., 2017; Zenke et al., 2017b; von Oswald et al., 2020; Liu & Liu, 2022; Raghavan & Balaprakash, 2021); and 4) modeling the parameter update uncertainty with Bayesian methods (Nguyen et al., 2018; Ebrahimi et al., 2020; Henning et al., 2021). In this paper, we focus on memory-replay-based methods since they often achieve SOTA performance. Memory-based methods include experience replay (Chaudhry et al., 2019b), which jointly trains the memory buffer data with current mini-batch. Meta Experience Replay (MER) (Riemer et al., 2019) adopts meta-learning to maximize transfer from previous examples and minimize interference. Hindsight Anchor Learning (HAL) (Chaudhry et al., 2021) using anchor points to mitigate forgetting on previous tasks. GEM (Lopez-Paz & Ranzato, 2017) and A-GEM (Chaudhry et al., 2019a) use the losses on the memory buffer data as inequality constraints, avoiding their increase but allowing their decrease to avoid forgetting. DER (Buzzega et al., 2020) further combines rehearsal with knowledge distillation.

## C.2 TASK-FREE CL

**Problem setup.** Task-free CL (He et al., 2019; Zeno et al., 2019; Aljundi et al., 2019b; Chrysakis & Moens, 2020; Lee et al., 2020) is a recent generalization of CL to the more complex cases, where data distribution shift could happen at any time during CL without explicit definition of tasks. A sequence of mini-batch labeled data $(\boldsymbol{x}_k, y_k, h_k)$ sequentially arrives at each timestamp $k$ and forms a non-stationary data stream; where $\boldsymbol{x}_k$ denotes the mini-batch data received at timestamp $k$, $y_k$ is the data label associated with $\boldsymbol{x}_k$, and $h_k$ is the hidden task identity associated with $\boldsymbol{x}_k$. During both the training and testing time, the task identity $h_k$ is not available to the learner. A more general definition of task-free CL in (Aljundi et al., 2019b) assumes no explicit partitions of tasks, and the data distribution can change arbitrarily. However, our proposed methods can be seamlessly applied to those more general scenarios.

**Existing work.** Most existing works in task-free CL are memory-replay-based methods (Chaudhry et al., 2019b;a). They directly perform memory replay on the raw data without any transformation. MIR (Aljundi et al., 2019a) proposes to replay the samples with which are most interfered. GEN-MIR (Aljundi et al., 2019a) further uses generative models to synthesize the memory examples. Gradient-based Sample Selection (GSS) (Aljundi et al., 2019c) focuses on *storing diverse examples* which is completely different from our method. Our works share a similar goal with GMED (Jin et al., 2021) which edits the memory buffer based on ADA, making the memory data harder but lacks diversity (Madry et al., 2018; Wang et al., 2021). There are several significant differences. First, GMED is a *gradient-based discrete step* memory editing method. Our memory transformer can obtain *infinite continuous time steps* transformation of memory data, which improves the memory diversity significantly. Second, GMED overwrites the memory data with the edited ones making the memory buffer data distribution significantly deviate from the original raw memory data distribution after many epochs of editing, especially in task-aware CL, which would decrease the performance. In contrast, our transformation process is reversible, i.e., the original raw data can be recovered from the transformed data. Thus, we do not need to overwrite the memory buffer data or keep an additional mini-batch transformed data with extra cost.

