# OpenReview forum: "Make Memory Buffer Stronger in Continual Learning: A Continuous Neural Transformation Approach"
_ICLR.cc/2023/Conference — Submitted to ICLR 2023_

### Official Review · Reviewer_A5Re · 2022-10-23

**Confidence:** 5
**Correctness:** 3
**Technical Novelty And Significance:** 2
**Empirical Novelty And Significance:** 3
**Recommendation:** 5

**Clarity, Quality, Novelty And Reproducibility:**

Clarity: overall the paper is clear.
Quality: the paper has some new ideas for continual learning but there are several confusing points.
Novelty: the novelty is fair.
Reproducibility: the paper doesn't seem easy to reproduce due to the introduction of many additional hyperparameters.

**Strength And Weaknesses:**

Strength:

1. The problem of preventing overfitting on memory is interesting and important for continual learning.
2. The idea of introducing continuous transformation to the memory is also interesting.


Weaknesses:

1. What's the computational complexity of the proposed approach? The transformations seem expensive which is infeasible in online continual learning. Also, the training of the memory transformer also takes time.

2. Section 3.4 is confusing. Are all the transformed data at different time stamps are combined? Will the transformed data be added to the memory buffer?

3. In ER, are any random augmentations applied to the memory data?

4. For ER, if the buffer size increases such that it consumes the same memory as the proposed method, how the performance will change? The proposed method introduces a memory transformer which takes more space than ER.

**Summary Of The Paper:**

In the paper, the authors propose a continuous and reversible memory transformation method to prevent overfitting on the memory in continual learning. The main idea is to increase the diversity of the data in the memory buffer while maintaining hardness. The authors propose Deterministic Continuous Memory Transformer to modify the data with infinite transformation functions to encourage diversity. The proposed method is compared with several existing memory editing methods. The results show the proposed method is better than existing ones.

**Summary Of The Review:**

The paper proposes to edit memory buffer with continuous transformation. However, there are several confusing pieces which make it hard to understand the paper. Also, the additional complexity is also an issue.

---

> ### Author Response · Authors · 2022-11-19
> **Response to Reviewer A5Re**
>
> Thanks for your helpful discussion!
>
> **Q1: What's the computational complexity of the proposed approach? The transformations seem expensive which is infeasible in online continual learning. Also, the training of the memory transformer also takes time.**
>
> **A1**: Thanks for pointing out this!  We analyze the complexity in appendix B.8. The computation complexity is related to the network size and integration time $T$. We use a very small memory transformer with a small number of parameters around 40K and also we control the $T$ to be small. In addition, we adopt backpropagation through the memory transformer for every 3 CL steps, and randomly sample $N$ points from the time interval, where $N \in [1, 5]$.  Those techniques also significantly improve the running efficiency.  By adopting these techniques, the running time is about 2.4 times of DER++.  As we mentioned in Appendix B.8, our method has similar complexity as GMED, which has about 2 times the complexity as DER++. It has been successfully applied in online continual learning.
>
> References:
>
> [1] Gradient-based Editing of Memory Examples for Online Task-free Continual Learning. NeurIPS 2021
>
> **Q2: Are all the transformed data at different time stamps are combined? Will the transformed data be added to the memory buffer?**
>
> **A2**:  We emphasize that our memory buffer capacity is the same as the baseline method. The transformed data at different time stamps are not combined.  The transformed memory data will not be added to the memory buffer. We first transform the data, then replay the transformed data, the transformed data will be discarded after use.  It is unnecessary to combine them together, or add them to the memory buffer. First, previous transformed data has already been learned by the CL learner. For new tasks, we need to transform the memory data adaptively.  Second,  combining them will increase a lot of memory storage cost. Third, our memory transformer is invertible, i.e., we can get the transformed data from memory buffer data. We can also recover original memory data from the transformed data.  We revised Section 3.5  （in blue color） to reflect this point.
>
> **Q3: In ER, are any random augmentations applied to the memory data?**
>
> **A3**: Yes, as we mentioned in Section 4.1 (Baseline), and we follow [1] and apply random crops and horizontal flips, etc, to both stream and buffer examples.
>
> Reference:
>
> [1] Dark Experience for General Continual Learning: a Strong, Simple Baseline. NeurIPS 2020
>
> **Q4: For ER, if the buffer size increases such that it consumes the same memory as the proposed method, how the performance will change? The proposed method introduces a memory transformer which takes more space than ER.**
>
> **A4**:  Our method DCMT only introduces about 30K additional parameters and SCMT only introduces about 40K parameters. For CIFAR100, each image is  $32\times32\times3$ (3072)  numbers, which is the same as ($\frac{40000}{3072} = 13$) images.    For Mini-ImageNet, each image is $84\times 84 \times 3$ (21168), which is the same memory cost of ($\frac{40000}{21168}=2$)  images.   We increase the memory buffer size for those datasets. We can observe that even with the same memory buffer size by increasing the baseline memory buffer size, our method still outperforms baseline by 2.9\% and 3.4\% on CIFAR100 and Mini-ImageNet for task-IL, respectively. We can observe similar improvement on class-IL as well.
>
>
> The performance by applying the proposed methods and baseline on DER++ on **CIFAR100** is .
> |  Method  | class-IL | task-IL |
> |---|---|---|
> |DER++ (memory 500)| $36.37\pm0.85$ | $75.64\pm0.60$ |
> |DER++(same memory as ours, 513)| $36.59 \pm 0.76$ | $ 75.81 \pm 0.54$ |
> |DER++DCMT | $38.68\pm0.81$ | $78.56\pm0.82$|
> |DER++SCMT | $38.56\pm0.93$ | $78.75\pm0.87$|
>
>
>  The performance by applying the proposed methods and baseline on DER++ on **Mini-ImageNet** is
> |  Method  | class-IL | task-IL |
> |---|---|---|
> |DER++ (memory 500)|$22.09\pm0.63$ | $61.26\pm0.57$ |
> |DER++(same memory as ours, 502)| $ 22.18\pm 0.72$ | $ 61.30\pm 0.67$ |
> |DER++ DCMT |  $24.53\pm0.89$|  $64.08\pm0.78$|
> |DER++ SCMT |  $24.81\pm0.97$ | $64.73\pm0.86$|
>
>
> **Q5:hyperparameters**
>
> **A5**: The main hyperparameters of our method are only $T$ and $\lambda$, we have provided detailed analysis in Appendix. So, the number of hyperparameters are quite few.

---

> ### Author Response · Authors · 2022-11-24
> **Follow up discussion with Reviewer A5Re**
>
> Dear Reviewer A5Re,
>
>             Thanks a lot for your helpful feedback!  We provided detailed response and revised our paper according to your suggestions.  We believe that our responses have addressed your concerns.  Could you let us know that are there any remaining concerns that need to be addressed?  If there are no unclear concerns, could you update your scores?  Thank you!
>
> Best,
>
> Authors

---

### Official Review · Reviewer_JHx6 · 2022-10-25

**Confidence:** 5
**Correctness:** 3
**Technical Novelty And Significance:** 3
**Empirical Novelty And Significance:** 2
**Recommendation:** 5

**Clarity, Quality, Novelty And Reproducibility:**

The paper is well written and easy to follow. Furthermore, the methods can be a novel approach for perturbing the memory data in exemplar memory.

**Strength And Weaknesses:**

**Pros:**

P1. The proposed method achieves state-of-the-art performance compared to the baselines, and using differential equation based perturbation in continual learning is a novel approach.

**Cons:**

C1. Though authors said the proposed methods can generate diverse examples, there is no quantitative results to verify this statement. It would be better to report the numbers using a metric such as precision-recall used in evaluating a generative models.

C2. The addition computation cost for performing single stochastic gradient step is not negligible. The proposed algorithm should execute the ODE or SDE solver and optimize the parameters for the generator during the learning process.

C3. There is no analysis on this method for verifying the proposed approach. For example, how much DCMT or SCMT affect the forgetting measure to the previous task? Since the memory data is perturbed, this rather produces negative impact on previous tasks.

**Summary Of The Paper:**

This paper proposed a novel method modifying the memory data in exemplar memory when performing experience replay in continual learning (CL). To address the problem of memory overfitting, the authors used deterministic and stochastic differential equations to perturb the memory data by increasing the cross-entropy loss to make the memory data hard to be classified. Furthermore, since differential equation based generative models are used, they can produce diverse memory data. In the experiments, the proposed methods achieves much higher accuracy than baselines.

**Summary Of The Review:**

I vote to reject this paper. Though the proposed methods can be novel, authors does not verified the effectiveness of DCMD or SCMT by analyzing the important components of proposed methods.

---

> ### Author Response · Authors · 2022-11-19
> **Response to Reviewer JHx6**
>
> Thank you for your valuable feedback!
>
> **Q1. Though authors said the proposed methods can generate diverse examples, there is no quantitative results to verify this statement. It would be better to report the numbers using a metric such as precision-recall used in evaluating a generative models.**
>
> **A1**:  Thanks for your suggestion!   We use the improved precision and recall used in generative models [1] to evaluate the diversity of the transformed data on CIFAR10.  (a) Precision measures the probability that a transformed image from a memory transformer falls within the support of original raw memory data. (b) Recall measures the probability that an original raw image from the memory buffer falls within the support of transformed memory data. The results are shown in the following table. The precision and recall are calculated with respect to the real original (no transformations) CIFAR10 data. Note that the goal of memory transformers is different from the goal of generative models. Since the goal of memory transformers is to transform the memory data to be a little bit different from original images. The goal of generative models is to generate images that are as close to the original images as possible.  The precision and recall drop (less than 1.0) indicates that the transformed memory data distribution slightly deviate from the original raw memory data distribution and the transformed memory data is more diverse.
>
> |  Method  | precision | recall |
> |---|---|---|
> |Real |1.0 |1.0|
> |DCMT |0.96  | 0.98|
> |SCMT |0.87 |0.94 |
>
> Reference:
>
> [1] Improved Precision and Recall Metric for Assessing Generative Models. NeurIPS 2019
>
> **Q2. The addition computation cost for performing single stochastic gradient step is not negligible. The proposed algorithm should execute the ODE or SDE solver and optimize the parameters for the generator during the learning process.**
>
> **A2**:  Thanks for your questions!  What we are saying is that the number of parameters of the memory transformer is negligible compared to that of backbone, ResNet18,  only accounting for less than 0.4% of the backbone. The proposed method indeed increases some computation cost, but is comparable compared to existing methods. We provided detailed computation cost analysis and efficiency improvement techniques in Appendix B.8 and also provided running time evaluation in Table 11. The computation cost depends on the network scale and integration time $T$, since we mentioned in page 8 that the amount of parameters is around 40K and we also controlled the integration time to be a small number. In addition, we adopt backpropagation through the memory transformer for every 3 steps, and randomly sample $N$ points from the time interval, where $N \in [1, 5]$.  Those techniques also significantly improve the running efficiency.  so the computation cost is not that high, it is about 2.4 times of DER++.   As we mentioned in Appendix B.8, our method has similar complexity as GMED[1], which is about 2 times the complexity as DER++. It has been successfully applied in online continual learning.
>
> References:
>
> [1] Gradient-based Editing of Memory Examples for Online Task-free Continual Learning. NeurIPS 2021
>
> **Q3. There is no analysis on this method for verifying the proposed approach. For example, how much DCMT or SCMT affect the forgetting measure to the previous task? Since the memory data is perturbed, this rather produces negative impact on previous tasks.**
>
> **A3**:  The memory data is perturbed, this does not necessarily lead to a negative impact on previous tasks. For example, GMED[1] and DER++[2] also perturb the memory data, but still provide a beneficial effect on previous tasks. In addition, we also provide backward transfer (BWT) in Table 6 Appendix B.5 to evaluate the forget measure on previous tasks. The results show that the BWT is better than the baseline method, indicating that the proposed method does not provide negative performance on previous tasks.
>
> References:
>
> [1] Gradient-based Editing of Memory Examples for Online Task-free Continual Learning. NeurIPS 2021
>
> [2]  Dark Experience for General Continual Learning: a Strong, Simple Baseline. NeurIPS 2020

---

> ### Author Response · Authors · 2022-11-24
> **Follow up discussion with Reviewer JHx6**
>
> Dear Reviewer JHx6,
>
>         Thank you again for your valuable comments!  We provided detailed response to address your questions.  We believe our responses have addressed all your concerns. Could you tell us that are there any remaining concerns that need to be clarified?  Thank you!
>
> Best,
>
> Authors

---

### Official Review · Reviewer_XsdS · 2022-11-04

**Confidence:** 4
**Correctness:** 3
**Technical Novelty And Significance:** 3
**Empirical Novelty And Significance:** Not applicable
**Recommendation:** 5

**Clarity, Quality, Novelty And Reproducibility:**

**Clarity:** While the writing is good, the paper's flow and clarity can be improved significantly.

**Novelty:** The idea is novel and original.

**Reproducibility:** Too many details are missing from the paper to make it reproducible.

**Quality:** The paper's quality can be improved (please refer to the weaknesses). I think the paper has a lot of potential, but in its current state, it feels a bit rushed.

**Strength And Weaknesses:**

## Strengths

* The paper addresses an interesting challenge and important problem with memory-replay-based approaches in CL.

* The proposed idea of using ODEs/SDEs for updating memory samples is novel and interesting.

* The empirical results of the paper indicate that DCMT/SCMT consistently improves the performance of memory-replay-based methods.

## Weaknesses

* The paper's flow could be significantly improved. For instance, Equation (11) seems to be the actual objective function used for the lower optimization problem proposed in Equation (9). It would be better to add the consistency loss directly to Equation (9) and then expand on why it is needed.

* Much of the presented details are unnecessary, while many critical details are completely missing from the paper. Most importantly, the velocity term $g(\cdot;t,\phi)$, which is at the heart of the proposed method, is never discussed in the paper (same goes for $\mu_\phi$ and $\sigma_\phi$). What classes of functions are used to update memory samples? From the results, it seems like $g(\cdot;t,\phi)$ acts on the color histograms, while one can use a large variety of transformations (e.g., the auto-augment approaches that also provide geometric transformations) as the parametric velocity term.

* The details of how the bilevel optimization problem is solved is entirely missing from the paper.

* The consistency loss in Equation (11) makes a lot of sense, and I believe it is critical for the success of the proposed methods. While the JS divergence regularizes for obtaining consistent labels, it does not guarantee it. Hence, if the regularization parameter $\lambda$ is not carefully selected, then one can end up with memory updates that cross the decision boundaries leading to harming the performance of previous tasks. The ablation study on $\lambda$ in Appendix B5 does not reflect this point, possibly due to the small range of $\lambda$ used in the ablation study.

* On Page 5, the authors state that even if  $g$ is not invertible they can recover $x(0)$ from $x(T)$. If I understand correctly, having $g$ we can perform forward propagation through time, i.e., naively $x(t+1)=x(t)+g(x(t);\phi)$ (or using Runge-Kutta as described in Algorithm 2 in the Appendix). However, one cannot retrieve $x(t)$ from $x(t+1)$, unless $g$ is invertible. Could you please elaborate on your statement?

**Summary Of The Paper:**

The paper focuses on memory-replay-based approaches for Continual Learning (CL). The authors justifiably argue that a continual learner could overfit to the memory buffer reducing the generalizability of the model on previously learned tasks. Hence, one could benefit from adjusting the samples in the memory to avoid such overfitting. The authors propose to update the samples in the memory buffer via first an Ordinary Differential Equation (ODE) approach, denoted as Deterministic Continuous Memory Transformer (DCMT), and second through a Stochastic Differential Equation (SDE) based approach, denoted as Stochastic Continuous Memory Transformer (SCMT). DCMT is identified via a parametric velocity/drift term $g(\cdot ; t, \phi)$, with parameters $\phi$, while SCMT uses both a drift and a diffusion term that are parameterized by $\phi$. Then, the authors propose a bilevel optimization where the lower level optimization finds the optimal parameters $\phi$ that provide memory updates that are adversarial/challenging (i.e., maximize the loss). At the same time, they maintain the class information (i.e., the CL provides consistent labels for the current and updated memory samples). Lastly, the authors show that DCMT and SCMT boost the performance of memory-replay-based CL approaches in task-aware (for task and class incremental learning) and task-free learning.


**Summary Of The Review:**

In summary, the paper addresses an important and interesting problem, and the proposed approach is original. However, many details are missing from the paper, and some critical points are completely ignored (please refer to weaknesses).

---

> ### Author Response · Authors · 2022-11-19
> **Response to Reviewer XsdS  (Part 1)**
>
> Thank you for your insightful and constructive feedback!
>
> **Q1: The paper's flow could be significantly improved. For instance, Equation (11) seems to be the actual objective function used for the lower optimization problem proposed in Equation (9). It would be better to add the consistency loss directly to Equation (9) and then expand on why it is needed.**
>
>
> **A1**: Thanks for your suggestions!  We revised our paper according to your suggestions. We changed the paper’s flow to reflect this.
>
>
> **Q2: Much of the presented details are unnecessary, while many critical details are completely missing from the paper. Most importantly, the velocity term g(⋅;t,ϕ), which is at the heart of the proposed method, is never discussed in the paper (same goes for μϕ and σϕ). What classes of functions are used to update memory samples? From the results, it seems like g(⋅;t,ϕ) acts on the color histograms, while one can use a large variety of transformations (e.g., the auto-augment approaches that also provide geometric transformations) as the parametric velocity term.**
>
> **A2**:  The transformation function g is a neural network that should satisfy that uniformly Lipschitz continuous in x and continuous in t.   A Lipschitz function is defined as
> if there exists a constant L > 0, such that |g(x) − g(y )| ≤ L|x − y | for all x, y ∈ E.
>
> Similarly, the $\mu_{\phi}$ and $\sigma_{\phi}$ also need to be Lipschitz functions.
> The above conditions are to make sure that the solution to an initial value problem of the differential equation exists and is unique [1]. Those conditions are very mild in existing literature as long as the network weights are finite, and use Lipshitz nonlinearities, e.g., tanh or relu. Our method directly applies the RGB value of input images, so the function classes are very flexible.
>
> Using more transformation, such as geometric transformations, e.g., in the auto-augment approaches, is definitely interesting.  Our proposed transformation is a completely learnable transformation that could adapt to each dataset automatically. In future works, we would explore the combination of our method with those geometric transformations as long as they are differentiable. We believe this should be a promising research direction.
>
>
> Reference:
>
> [1] Earl A Coddington and Norman Levinson. Theory of ordinary differential equations. Tata McGrawHill Education, 1955.
>
> **Q3: The details of how the bilevel optimization problem is solved is entirely missing from the paper.**
>
> **A3**: Thanks for your suggestions! We provide more details in Algorithm 1 for solving the bi-level optimization and more details in Appendix B.3 for the gradient calculation of the memory transformer.
>
> **Q4: The consistency loss in Equation (11) makes a lot of sense, and I believe it is critical for the success of the proposed methods. While the JS divergence regularizes for obtaining consistent labels, it does not guarantee it. Hence, if the regularization parameter λ is not carefully selected, then one can end up with memory updates that cross the decision boundaries leading to harming the performance of previous tasks. The ablation study on λ in Appendix B5 does not reflect this point, possibly due to the small range of λ used in the ablation study.**
>
> **A4**:  For the hyperparameter $\lambda$,  (1) when the $\lambda$ becomes larger, the regularization strength becomes stronger, thus the labels could become more consistent for transformed data and original raw data. But if the regularization weight is too large, it will restrict the neural network output flexibility, may decrease the performance.     (2)When the  $\lambda$ becomes smaller, the regularization strengths become weaker, the labels may become less consistent, and may decrease the performance. The following experiments with DCMT on more varied hyperparameters show the results.
> |  Dataset  | $\lambda=0.0$ |$\lambda=0.2$  |$\lambda=1.0$  | $\lambda=3.0$  | $\lambda=5.0$ |$\lambda=15.0$ |
> |---|---|---|---|---|---|---|
> |CIFAR10|  $94.62\pm 0.49$  | $94.73\pm 0.62$ |$94.98\pm 0.54$ | $95.29\pm 0.54$ | $94.85\pm 0.57$| $94.49\pm 0.68$||
> |MiniImagenet | $63.71\pm 0.72$ | $63.93\pm 0.67$ |$64.37\pm 0.53$ | $64.08\pm 0.78$ | $64.11\pm 0.73$|$63.53\pm 0.75$|

---

> > ### Author Response · Authors · 2022-11-19
> > **Response to Reviewer XsdS (Part 2)**
> >
> > **Q5: On Page 5, the authors state that even if g  is not invertible they can recover x(0)  from  x(T). If I understand correctly, having g  we can perform forward propagation through time, i.e., naively x(t+1)=x(t)+g(x(t);ϕ) (or using Runge-Kutta as described in Algorithm 2 in the Appendix). However, one cannot retrieve x(t) from  x(t+1), unless
> > g is invertible. Could you please elaborate on your statement?**
> >
> > **A5**: Thanks for your question!
> > We first derive why the transformation is invertible and why the function g does not need to be invertible.
> > The forward integration on the interval $[0, T]$ is as the following:
> >
> > $x^m (T) = x^m(0)+ \int_{0}^{T} g(x^m(t), t, \phi) dt $
> >
> > We then multiple $-1$ for both sides of the above equations, obtain the following equation:
> >
> > $-x^m (T) = -x^m(0) -\int_{0}^{T} g(x^m(t), t, \phi) dt $
> >
> >  We then rearrange the above equation as following:
> >
> > $x^m(0)  = x^m (T)  -\int_{0}^{T} g(x^m(t), t, \phi) dt $.
> >
> > Then due to the following property of integration
> >
> > $ -\int_{0}^{T} g(x^m(t), t, \phi) dt = \int_{T}^{0} g(x^m(t), t, \phi) dt$   (reverse the integration interval to be $[T, 0]$, the integration value becomes negative)
> >
> > We can obtain the following  reverse integration:
> >
> > $x^m(0) = x^m(T) + \int_{T}^{0} g(x^m(t), t, \phi) dt$
> >
> > While we can observe that in the above derivation, there is no restriction that $g$ should be invertible.
> >
> >
> >
> > In terms of numerical integration, we did not  invert the iterative forward integration process. It is unnecessary to directly invert the numerical iteration, but instead following the standard integration scheme (e.g., Runge-Kutta method ) on the reverse integration interval $[T, 0]$. It is worth noting that the forward integration time interval is $[0, T]$. That is, if we integrate the above reverse integration in the interval $[T, 0]$, based on the property of integration
> >
> > $ -\int_{0}^{T} g(x^m(t), t, \phi) dt = \int_{T}^{0} g(x^m(t), t, \phi) dt$.
> >
> > We would recover from $x(0)$ from $x(T)$ and do not need to assume $g$ is invertible.

---

> ### Author Response · Authors · 2022-11-24
> **Follow up discussion with Reviewer XsdS**
>
> Dear Reviewer XsdS,
>
>             Thank you so much for your detailed comments and suggestions!  We provided detailed response to your concerns and hope our response addressed your concerns.  Could you let us know that are there any remaining concerns?  We could provide further clarifications.  Thank you!
>
>
> Best,
>
> Authors

---

### Official Review · Reviewer_vLJC · 2022-11-04

**Confidence:** 3
**Correctness:** 2
**Technical Novelty And Significance:** 2
**Empirical Novelty And Significance:** 2
**Recommendation:** 5

**Clarity, Quality, Novelty And Reproducibility:**

This paper proposes an interesting idea of using diffential equations to parameterize the continuous transformation for augmenting the memory buffer. The overall idea is interesting, but more justifications and evaluations are needed in order to show the effectiveness of the proposed method.


**Strength And Weaknesses:**

Strength:

- The idea of augmenting the memory buffer with a transformation is straightforward and interesting.

- The paper is overall easy to follow.

Weakness:

- Why using differential equations to model the memory transformation is poorly motivatied and also lack of justificaitons.

- Despite the gain of the method, there is more must-to-compare baselines. For example, how about using a standard neural network to serve as the transformation, or use standard noise to augment the memory, similar to [Prototype augmentation and selfsupervision for incremental learning. In CVPR, 2021].

**Summary Of The Paper:**

This paper proposes a method to augment the memory buffer, aiming to increase the data diversity. Specifically, this paper chooses to use differential equations to model the memory transformation. The experimental results on standard continual learning benchmarks show some improvements of the proposed method.

**Summary Of The Review:**

Overall, I find this paper below the bar of acceptance, especially because of the lack of baselines (other perturbations for the memory buffer). It requires more ablation study and more justifications for the usage of differntial equation modeling.

---

> ### Author Response · Authors · 2022-11-19
> **Response to Reviewer vLJC**
>
> Thank you for your thoughtful comments!
>
> **Q1: Why using differential equations to model the memory transformation is poorly motivated and also lack of justifications.**
>
> **A1**:  Thanks for pointing out this! Please refer to our updated paper version in Section 3.2  (A preliminary approach to increase memory diversity)  in blue color for detailed motivation explanations.
>
>
> **Q2: there is more must-to-compare baselines. For example, how about using a standard neural network to serve as the transformation, or use standard noise to augment the memory, similar to [Prototype augmentation and self-supervision for incremental learning. In CVPR, 2021].**
>
> **A2**: Thanks for pointing out these baselines!
>
> * For using a neural network to do transformation, we provide why do we need to use continuous differential equations instead of standard neural networks in our updated paper version in Section 3.2  (A preliminary approach to increase memory diversity)  in blue color for detailed motivation explanations. Please refer to the details there.
>
> * For adding random gaussian noise to the memory data similar to [1], we also provide additional comparisons in the following experiments on CIFAR10, CIFAR100, Mini-ImageNet in Table 1 (Section 4.1 (Experiment) in the main text revision), please refer to the updated version.  Adding noise only brings marginal gains or even worse performance since adding noise only adds a simple transformation pattern, but still lacks diversity. When training for a long time, the network could also memorize the noise pattern, and still overfit. In contrast, our method adds more diversity by directly transforming the pixel values with expressive transformation function.
>
> * Our method also outperforms neural network transformation in Section 4.1 since discrete step transformation by neural network can be viewed as discrete approximation of our continuous transformation. The performance of using a standard neural network (We name this our proposed preliminary approach as **discrete transformation (DT)**) is upper bounded by our continuous transformation. Also, DT needs to store all the intermediate transformations. In contrast, DCMT and SCMT do not need to store them. Thus, DCMT and SCMT is much more memory efficient than DT.
>
>
> The following tables shows the results on two datasets with memory size of 500 by adding random noise to the memory buffer or using neural network transformation to transform the memory data.
>
> Results on **CIFAR100** by comparing to adding random gaussian noise and with standard neural network transformation
>
>
> |  Method  |  Class-IL  | Task-IL |
> |---|---|---|
> |DER++| $36.37\pm0.85$ |$75.64\pm0.60$|
> |DER++noise | $36.02\pm0.91$ | $75.38\pm 0.68$ |
> |DER++ DT(neural network)| $37.53\pm 0.97$ |$77.21\pm 0.79$|
> |DER++DCMT   | $38.68\pm0.81$  |$78.56\pm0.82$|
> |DER++SCMT  | $38.56\pm0.93$  |$78.75\pm0.87$|
>
>
> Results on **Mini-ImageNet** by comparing to adding random gaussian noise and with standard neural network transformation
>
> |  Method  |  Class-IL  | Task-IL |
> |---|---|---|
> |DER++| $22.09\pm0.63$ | $61.26\pm0.57$|
> |DER++noise | $22.32\pm0.87$ | $61.51\pm0.71$|
> |DER++DT(neural network)|  $23.45\pm0.93$| $63.03\pm0.75$|
> |DER++DCMT   | $24.53\pm0.89$| $64.08\pm0.78$|
> |DER++SCMT  | $24.81\pm0.97$ | $64.73\pm0.86$|
>
>
> References:
> [1]: Prototype augmentation and self-supervision for incremental learning. In CVPR, 2021

---

> > ### Comment · Reviewer_vLJC · 2022-11-22
> > **Response**
> >
> > Thanks for the response. The empirical results look okay, but the proposed one is not as signficant as before.
> >
> > I am still leaning towards rejection due to the lack of motivation for the complex method to model the memory transformation. I am confused by why we need to model memory tranformation using the continuous dynamic system. I fail to see the necessity of using this for augmenting the memory buffer. If you want more diverse memory buffer, why not learn the class distribution instead?
> >
> > Given the above reason, I stick to my previous rating.

---

> > > ### Author Response · Authors · 2022-11-22
> > > **Response to Reviewer vLJC**
> > >
> > > * Thank you for your reply! We would like to emphasize that the method **discrete transformation (DT) in Section 3.2** , **DCMT** and **SCMT** are all our proposed methods. We propose * **DT** to show why we need continuous transformation instead of discrete transformation using neural network*.  DCMT and SCMT outperform DT in terms of various performance evaluation metrics in CL, but also they have significant memory storage advantage, i.e., $\mathcal{O}(1)$,  compared to DT,  whose memory cost scales with $\mathcal{O}(n)$; where $n$ is the number of transformation steps. This advantage becomes more obvious when $n$ is large as we mentioned and elaborated in details in Section 3.2
> > >
> > > * *If you want more diverse memory buffer, why not learn the class distribution instead?*. Could you elaborate more in details how to estimate the class information to generate more diverse memory data? Thank you!

---

> > > > ### Comment · Reviewer_vLJC · 2022-12-08
> > > > **Lack of justification for the complex modeling of buffer**
> > > >
> > > > I have read the response and I appreciate the authors' efforts. To conclude, I am still not convinced by the response and decide to stick to my current rating.
> > > >
> > > > - Specifically, I still didn't get the benefits of using a continuous depth network (say neural ODE). I can see that it is more parameter-efficient at the cost of being less flexible (less model capacity). The only justification I see is from the empirical side, but the experimental results could vary a lot with different settings and different hyperparameters (and the continuous transformation also has more hyperparameters, such as the stopping depth. Even for stochastic depth, how you sample it is also a hyper-parameter choice). In general, empirical justification could not be the determining reason that the continuous transformation is indeed superior.
> > > >
> > > > - I think from a high-level perspective, that the proposed method is poorly motivated, because there are a number of ways to increase diversity in the memory buffer. Just to name a few: adding noise to the input of the network, doing dropout to generate perturbed transformation (just like SimCSE [https://arxiv.org/abs/2104.08821]) and even modeling the class distribution using some generative models (such as generative memory replay [https://arxiv.org/abs/1705.08690]).
> > > >
> > > > - I didn't really find this paper inspiring, since the proposed method simply try a few different parameterizations for the transformation neural networks. Even from a purely empirical perspective, the performance gain compared to DER++ is also not signficant enough on CIFAR-100.

---

> > > > > ### Author Response · Authors · 2022-12-10
> > > > > **Further Response to Reviewer vLJC**
> > > > >
> > > > > Dear Reviewer vLJC:
> > > > >
> > > > > We thank you for your reply!  We will clarify your questions in the following:
> > > > >
> > > > > **Q: Specifically, I still didn't get the benefits of using a continuous depth network (say neural ODE).**
> > > > >
> > > > > **A**: We explained the advantage in our previous response. We would like to emphasize that we have explained in Section 3.2 why we need continuous transformation instead of discrete transformation using neural network.  **DCMT** and **SCMT** outperform DT in terms of significant memory storage advantage, i.e., $\mathcal{O}(1)$,  compared to DT,  whose memory cost scales with $\mathcal{O}(n)$; where $n$ is the number of transformation steps. The advantage of using continuous transformation is **not only parameter efficient, but also we do not need to store all the intermediate features as discrete layers**. Since the CL mainly deals with the scenario where the memory buffer is small. Memory efficiency is critical for CL algorithms. This advantage becomes more obvious when $n$ is large as we mentioned and elaborated in details in Section 3.2.
> > > > >
> > > > >
> > > > > **Q: adding noise to the input of the network**
> > > > >
> > > > > **A**: For adding noise to the input of the network,  we have explained in our previous response.
> > > > > For adding random gaussian noise to the memory data, we also provide additional comparisons in the following experiments on CIFAR10, CIFAR100, Mini-ImageNet in Table 1 (Section 4.1 (Experiment) in the main text revision), please refer to the updated version. Adding noise only brings marginal gains or even worse performance since adding noise only adds a simple transformation pattern, but still lacks diversity. When training for a long time, the network could also memorize the noise pattern, and still overfit. In contrast, our method adds more diversity by directly transforming the pixel values with an expressive transformation function.
> > > > >
> > > > >
> > > > > Results on **CIFAR100** by comparing to adding random gaussian noise and with standard neural network transformation
> > > > >
> > > > >
> > > > > |  Method  |  Class-IL  | Task-IL |
> > > > > |---|---|---|
> > > > > |DER++| $36.37\pm0.85$ |$75.64\pm0.60$|
> > > > > |DER++noise | $36.02\pm0.91$ | $75.38\pm 0.68$ |
> > > > > |DER++ DT(neural network)| $37.53\pm 0.97$ |$77.21\pm 0.79$|
> > > > > |DER++DCMT   | $38.68\pm0.81$  |$78.56\pm0.82$|
> > > > > |DER++SCMT  | $38.56\pm0.93$  |$78.75\pm0.87$|
> > > > >
> > > > >
> > > > > Results on **Mini-ImageNet** by comparing to adding random gaussian noise and with standard neural network transformation
> > > > >
> > > > > |  Method  |  Class-IL  | Task-IL |
> > > > > |---|---|---|
> > > > > |DER++| $22.09\pm0.63$ | $61.26\pm0.57$|
> > > > > |DER++noise | $22.32\pm0.87$ | $61.51\pm0.71$|
> > > > > |DER++DT(neural network)|  $23.45\pm0.93$| $63.03\pm0.75$|
> > > > > |DER++DCMT   | $24.53\pm0.89$| $64.08\pm0.78$|
> > > > > |DER++SCMT  | $24.81\pm0.97$ | $64.73\pm0.86$|
> > > > >
> > > > >
> > > > >
> > > > > **Q: doing dropout to generate perturbed transformation.**
> > > > >
> > > > >
> > > > > **A**: Our method is  *orthogonal* to the dropout since dropout is done on the CL method parameters. Our method is from a memory *data perspective*. The experiments are as following:
> > > > >
> > > > > |  Method  |  Class-IL  | Task-IL |
> > > > > |---|---|---|
> > > > > |DER++| $22.09\pm0.63$ | $61.26\pm0.57$|
> > > > > |DER++noise | $22.32\pm0.87$ | $61.51\pm0.71$|
> > > > > |DER++Dropout|  $22.50\pm 1.06$| $62.16\pm 0.93$|
> > > > > |DER++DCMT   | $24.53\pm0.89$| $64.08\pm0.78$|
> > > > > |DER++SCMT  | $24.81\pm0.97$ | $64.73\pm0.86$|
> > > > >
> > > > > This shows the effect of dropout is limited.
> > > > >
> > > > >
> > > > >
> > > > > **Q: even modeling the class distribution using some generative models (such as generative memory replay [https://arxiv.org/abs/1705.08690]).**
> > > > >
> > > > > **A**: We have explained in our previous response. **Since the memory buffer is usually small (e.g., 500), the generative model cannot estimate the memory distribution accurately since the generative model needs a large amount of data to do accurate estimation.**
> > > > >
> > > > > Results on **CIFAR100** with memory size 500
> > > > >
> > > > > |  Method  |  Class-IL  | Task-IL |
> > > > > |---|---|---|
> > > > > |DER++| $36.37\pm0.85$ |$75.64\pm0.60$|
> > > > > |DER++noise | $36.02\pm0.91$ | $75.38\pm 0.68$ |
> > > > > |DER++class distribution | $33.16\pm0.86$ | $70.29\pm 0.98$ |
> > > > > |DER++ DT(neural network)| $37.53\pm 0.97$ |$77.21\pm 0.79$|
> > > > > |DER++DCMT   | $38.68\pm0.81$  |$78.56\pm0.82$|
> > > > > |DER++SCMT  | $38.56\pm0.93$  |$78.75\pm0.87$|
> > > > >
> > > > >
> > > > > The results indicate that estimating the class distribution with generative models performs poorly. This is due to the fact that the **generative model needs a large amount of data to perform well, but the memory buffer size is usually small. Estimating the class distribution with memory size of 500 is hard to be accurate**. Thus, the performance by estimating the class distribution even underperforms all the other methods.
> > > > >
> > > > > **Q: the performance gain compared to DER++ is also not signficant**
> > > > >
> > > > > **A**: For the performance gains, our methods outperforms DER++ on CIFAR100 by 3.1% on task incremental learning, and 2.3% on class incremental learning. Note that DER++ is already a very strong baseline in CL. Furthermore, class incremental learning is a challenging task and improvement with 2.3% is already a significant improvement. Our method also outperform the DER++ more than 3% on mini-ImageNet.

---

> > > ### Author Response · Authors · 2022-11-23
> > > **Response to Reviewer vLJC**
> > >
> > > We further perform extra experiment with generative adversarial networks  on **CIFAR100** to estimate the class distribution. At each iteration, we randomly sample data from the generative models for replay. The results on CIFAR100 are as following:
> > >
> > > Results on **CIFAR100** with memory size 500
> > >
> > >
> > > |  Method  |  Class-IL  | Task-IL |
> > > |---|---|---|
> > > |DER++| $36.37\pm0.85$ |$75.64\pm0.60$|
> > > |DER++noise | $36.02\pm0.91$ | $75.38\pm 0.68$ |
> > > |DER++class distribution | $33.16\pm0.86$ | $70.29\pm 0.98$ |
> > > |DER++ DT(neural network)| $37.53\pm 0.97$ |$77.21\pm 0.79$|
> > > |DER++DCMT   | $38.68\pm0.81$  |$78.56\pm0.82$|
> > > |DER++SCMT  | $38.56\pm0.93$  |$78.75\pm0.87$|
> > >
> > >
> > > The results indicate that estimating the class distribution with generative models perform poorly. This is due to the fact that generative model needs large amount of data to perform well, but the memory buffer size is usually small. Estimating the class distribution with memory size of 500 is hard to be accurate. Thus, the performance by estimating the class distribution even underperform all the other methods.

---

> > > ### Author Response · Authors · 2022-11-24
> > > **Follow up discussion with Reviewer vLJC**
> > >
> > > Dear reviewer vLJC,
> > >
> > >           We  thank you for your thoughtful comments again!   We provided additional detailed response to your concerns, and believe that we have addressed your remaining concerns.  We would like to ask that is there anything else that needs us to further clarify? We are very glad to answer them. If there are no unclear concerns, could you update your score? Thank you!
> > >
> > > Best,
> > >
> > > Authors

---

### Author Response · Authors · 2022-11-19
**Revision Summary**

Revision summary:

The revisions in the main text and Appendix  are shown in blue color.

* We add section 3.2 to reflect why we adopt continuous memory transformation instead of standard neural network

* We add more experiments compared to adding random noise and using neural network transformation

* We add more descriptions on the memory transformation function classes and bi-level optimization details

* We add more memory maintenance details

* We move part of related works and part of method description  in Appendix due to space limitations.

---

### Author Response · Authors · 2022-12-08
**Request Feedback**

Dear Reviewers,

             Thank you all for your helpful comments and constructive feedback!  We have tried our best to address your concerns and we believe we have addressed your concerns. Since the discussion period is close soon, could you please provide us further feedback about which concerns have not been addressed?  If your concerns have been addressed, could you please update your scores to reflect our update?  Thank you!

Best,
Authors

---

### Decision · Program_Chairs · 2023-01-20

**Decision:**

Reject

**Justification For Why Not Higher Score:**

The paper lack clarity about the proposed method and motivation for introducing such a complex approach.


**Justification For Why Not Lower Score:**

N/A

**Metareview: Summary, Strengths And Weaknesses:**

Memory buffers are often used for the continual learning framework to avoid forgetting previous tasks. However, the continual learning models tend to overfit to these memory buffers which cancels out their benefits. To mitigate this issue, the paper introduces a method for learning continuous learnable transformations of the data that enriches the memory buffer and favors its diversity thus avoiding memory overfitting.

Below is a summary of the main concerns of the reviewers which the AC also shares:
The paper lacks motivation for the proposed method, in particular the usage of continuous-time dynamics for modeling the replay-buffer, which adds a lot of complexity to the method without a clear benefit. The authors added some clarifications stating that ODE/SDEs allow for infinitely many memory samples. However, this does not really explain why it is better than simpler approaches, such as using a conditional generative model without continuous-time dynamics.
Clarity: most reviewers agree that the complexity of the method and the presentation makes the paper hard to understand. The authors did not provide complete details for the algorithms as pointed out by Reviewer XsdS. Despite the additional clarifications during the rebuttal, it is still unclear whether the algorithm accounts for the dependence of the optimal memory buffer of the parameters of the CL model as it should be for bilevel optimization problems. This can pose a significant challenge especially since the algorithm would have to backpropagate through the ODE/SDE. In the case where the authors do not consider such dependence, then the problem reduces to a game and not a bilevel problem, hence the problem formulation must be changed.

While the idea is promising and novel, the paper suffers from a number of limitations raised by the reviewers.